# Structural bases for aspartate recognition and polymerization efficiency of cyanobacterial cyanophycin synthetase

Takuya Miyakawa [1,7,8], Jian Yang [1,2,3,8], Masato Kawasaki[4,5,8], Naruhiko Adachi [4,8], Ayumu Fujii[1], Yumiko Miyauchi[1], Tomonari Muramatsu[1], Toshio Moriya [4], Toshiya Senda [4,5,6] ✉ & Masaru Tanokura [1] ✉

Cyanophycin is a natural biopolymer consisting of equimolar amounts of aspartate and arginine as the backbone and branched sidechain, respectively. It is produced by a single enzyme, cyanophycin synthetase (CphA1), and accumulates as a nitrogen reservoir during $N_2$ fixation by most cyanobacteria. A recent structural study showed that three constituent domains of CphA1 function as two distinct catalytic sites and an oligomerization interface in cyanophycin synthesis. However, it remains unclear how the ATP-dependent addition of aspartate to cyanophycin is initiated at the catalytic site of the glutathione synthetase-like domain. Here, we report the cryogenic electron microscopy structures of CphA1, including a complex with aspartate, cyanophycin primer peptide, and ATP analog. These structures reveal the aspartate binding mode and phosphate-binding loop movement to the active site required for the reaction. Furthermore, structural and mutational data show a potential role of protein dynamics in the catalytic efficiency of the arginine condensation reaction.

Bacteria efficiently convert carbon and nitrogen sources into various classes of intracellular and extracellular biopolymers with distinct chemical properties and biological functions[1,2]. These polymeric substances provide energy storage, adhesion, or protection functions in cells, and their production is regulated by environmental stimuli[3]. Over the past few decades, a better understanding of the molecular mechanism underlying the biosynthesis of biopolymers has shed light on the production of tailor-made biopolymers and their utility in the industrial field[4,5]. Among them, polypeptides contribute significantly to a sustainable society due to their biomass origin, biodegradability, and unique functionality[6].

Cyanophycin, a nonribosomally synthesized polypeptide consisting of equimolar amounts of aspartate and arginine as the backbone and branched sidechain, respectively, was first identified in cyanobacteria in the form of opaque and light-scattering cytoplasmic granules (Supplementary Fig. 1). Cyanophycin has high nitrogen to carbon ratio of 1:2, making it a reservoir of fixed nitrogen in most cyanobacteria[7]. Simultaneous oxygenic photosynthesis and $N_2$ fixation

[1]Department of Applied Biological Chemistry, Graduate School of Agricultural and Life Sciences, The University of Tokyo, 1-1-1 Yayoi, Bunkyo-ku, Tokyo 113-8657, Japan. [2]CAS Key Laboratory of Tropical Marine Bio-resources and Ecology, Guangdong Key Laboratory of Marine Materia Medica, South China Sea Institute of Oceanology, Chinese Academy of Sciences, Guangzhou, China. [3]Southern Marine Science and Engineering Guangdong Laboratory (Guangzhou), Guangzhou, China. [4]Structural Biology Research Center, Photon Factory, Institute of Materials Structure Science, High Energy Accelerator Research Organization (KEK), 1-1 Oho, Tsukuba, Ibaraki 305-0801, Japan. [5]Department of Materials Structure Science, School of High Energy Accelerator Science, The Graduate University of Advanced Studies (Soken-dai), 1-1 Oho, Tsukuba, Ibaraki 305-0801, Japan. [6]Faculty of Pure and Applied Sciences, University of Tsukuba, 1-1-1 Tennodai, Ibaraki 305-8571, Japan. [7]Present address: Division of Integrated Life Science, Graduate School of Biostudies, Kyoto University, Kitashirakawa-oiwakecho, Sakyo-ku, Kyoto 606-8502, Japan. [8]These authors contributed equally: Takuya Miyakawa, Jian Yang, Masato Kawasaki, Naruhiko Adachi. ✉e-mail: toshiya.senda@kek.jp; amtanok@mail.ecc.u-tokyo.ac.jp

are a significant challenge for microorganisms because the $O_2$ produced from $CO_2$ fixation is inhibitory to nitrogenase, which catalyzes the conversion of $N_2$ to $NH_3$[8]. To solve this problem, diazotrophs have developed physical strategies to temporally and spatially separate nitrogenase from $O_2$. The heterocyst-forming cyanobacteria, such as *Anabaena* spp., perform $CO_2$ fixation and $N_2$ fixation in different cells: $CO_2$ fixation in the vegetative cells and $N_2$ fixation in heterocysts[9]. Other cyanobacteria, such as *Trichodesmium* spp., temporally segregate the processes by a diel cycle with $CO_2$ fixation during the day and $N_2$ fixation at night[10]. The fixed nitrogen accumulates at night as cyanophycin granules separated from the other cellular components but not as compounds that affect cellular metabolic dynamics. Cyanophycin is degraded by cyanophycinase and isopeptidase to release nitrogen to support cell growth during the day. Thus, cyanophycin granules serve as dynamic storage bodies in diazotrophic cyanobacteria to uncouple $N_2$ fixation from overall growth dynamics[9]. Cyanophycin accumulation enables the optimization of the nitrogen utilization of cyanobacteria in natural environments with fluctuating and limiting nitrogen supplies[11].

The biosynthesis of cyanophycin is achieved by a single enzyme named cyanophycin synthetase (CphA1) that catalyzes peptide elongation coupled with ATP hydrolysis[5]. CphA1 uses a low-molecular-weight cyanophycin (at least 3–4 dipeptides long) as a primer for cyanophycin polymerization[12]. Recently, the tertiary structure of CphA1 has become available and reveals that amino acid polymerization is driven by three distinct domains[13]: the N-terminal domain (N domain), the middle glutathione synthetase-like domain (G domain) with the ATP-grasp fold, and the MurE-like muramyl ligase domain (M domain). The discrete active sites of the G and M domains are used for extending the Asp backbone and attaching the Arg sidechain, respectively (Supplementary Fig. 1). The N domain has been proposed to anchor cyanophycin polymers through electrostatic interactions[13]. However, it remains unclear how ATP-dependent addition of aspartate to cyanophycin is initiated at the catalytic site of the G domain.

In this work, given the importance of CphA1 in cyanophycin biosynthesis and the biotechnological value that structural information on enzyme-substrate interactions would provide, we investigate the CphA1 enzyme from the marine diazotroph *Trichodesmium erythraeum* IMS101 (*Te*CphA1), an important contributor to global nitrogen and carbon cycling[14]. We solve the *Te*CphA1 structure bound to ATP analogs, cyanophycin primer peptides, and the substrate aspartate, in addition to apo and ATP analog-bound structures. Key elements, including residues and protein dynamics required for enzymatic activity, are identified based on experimental evidence of the G and M domains. In particular, we reveal the structural bases for aspartate recognition and condensation, which deepen our understanding of the reaction of cyanophycin biosynthesis in the G domain. Moreover, our data show a potential role of protein dynamics in the catalytic efficiency for the reaction of Arg sidechain condensation.

## Results

### Cryo-EM structures of *Te*CphA1
The *Te*CphA1 structures in three different states were determined by single-particle cryogenic electron microscopy (cryo-EM): apo, the adenosine-5′-(γ-thio)-triphosphate (ATPγS, an ATP analog)-bound state, and the substrate-bound state (*Te*CphA1-aspartate-ATPγS-cyanophycin primer peptide complex) at 2.91, 2.96, and 2.64 Å resolution, respectively (Fig. 1a, Supplementary Figs. 2–5, and Supplementary Table 1). The cryo-EM data of the substrate-bound state were collected under a high concentration of L-aspartate (100 mM), cyanophycin primer peptide, (β-Asp-Arg)$_4$, and ATPγS. *Te*CphA1 consists of three domains: an N-terminal domain (N domain, residues 1–161) that adopts a partially similar structure to *E. coli* RNA-polymerase α-subunit[13], a glutathione synthetase-like domain (G domain, residues 162–487), and a MurE-like muramyl ligase domain (M domain, residues

488–876) (Fig. 1b). *Te*CphA1 basically adopts a tetrameric assembly with $D_2$ symmetry (Fig. 1a and Supplementary Fig. 6), which is a common architecture with another cyanobacterial CphA1 from *Synechocystis* sp. UTEX2470 (*Su*CphA1) (Supplementary Fig. 7)[13].

The G domain, which is responsible for backbone elongation through linking aspartate to the C-terminal backbone carboxy group of cyanophycin, is based on $G_{core}$ (residues 162–234, 306–324, and 400–487), with $G_{lid}$ (residues 235–305) and $G_\omega$ (residues 325–399) modules inserted into the sequence (Fig. 1b and Supplementary Fig. 1). $G_{core}$ is a central module that provides a large dimer interface with another $G_{core}$ in the *Te*CphA1 tetramer (Fig. 1c and Supplementary Fig. 8a). $G_{lid}$ acts as a lid of the ATP-binding site of $G_{core}$, and $G_\omega$ is positioned to surround the substrate-binding site together with $G_{lid}$. The relative map intensity of $G_\omega$ versus $G_{core}$ and $G_{lid}$ is higher in the substrate-bound state than in the other two states (Fig. 1a), suggesting that $G_\omega$ is clearly visible as a result of reduced mobility, which is probably due to the binding of aspartate and (β-Asp-Arg)$_4$.

The M domain, which catalyzes the addition of the Arg sidechain to the β-carboxy group of the C-terminal Asp residue of cyanophycin (Fig. 1b), is divided into two modules, $M_{core}$ (residues 488–723) and $M_{lid}$ (residues 724–876). $M_{core}$ contacts the intramolecular N domain and three $G_{core}$ to form the *Te*CphA1 tetramer (Fig. 1c and Supplementary Fig. 8b), resulting in the clear density map of $M_{core}$ (Fig. 1a), and ATPγS binds to the surface pocket of $M_{core}$. Therefore, $M_{core}$ assumes a distinct catalytic site from the G domain in *Te*CphA1, which is similar to the structural observation of *Su*CphA1[13]. On the other hand, the structural evidence of the ATP-bound forms of the G and M domains remains insufficient for the CphA1 enzymes from *Acinetobacter baylyi* DSM587 (*Ab*CphA1) and *Tatumella morbirosei* DSM23827 (*Tm*CphA1)[13]. $M_{lid}$ is not well resolved in the apo- and ATPγS-bound states (Fig. 1a and Supplementary Fig. 8c), suggesting the mobile nature of $M_{lid}$. However, in the substrate-bound state, the $M_{lid}$ modules were partially resolved around ATPγS in chains A and B (Fig. 1a and Supplementary Fig. 6). The density of $M_{lid}$ was relatively weak compared with that of $M_{core}$. A large shift of $M_{lid}$ was observed in the other CphA1 enzymes[13] and Mur ligases[15–17].

*Te*CphA1 has a high sequence identity (69.6%) and similarity (83.2%) to *Su*CphA1 in the region consisting of the N, G, and M domains (residues 1–876). We compared the structures of *Te*CphA1 in the substrate-bound state and *Su*CphA1 that bound a cyanophycin analog (β-Asp-Arg)$_8$-NH$_2$ and an ATP analog 5′-adenylylmethylenediphosphonate (AMPPCP) to the G domain (PDB 7LGJ)[13]. These structures are similar to each other. The overall structures of the N domain and each module of the G and M domains ($G_{core}$, $G_{lid}$, $G_\omega$, $M_{core}$, and $M_{lid}$) were quite similar between *Te*CphA1 and *Su*CphA1 (root-mean-square deviation (RMSD) = 0.521–1.601 Å). The spatial arrangement of the N domain and the $G_{core}$ and $M_{core}$ modules was also well conserved between *Te*CphA1 and *Su*CphA1 (Supplementary Fig. 7b), whereas the relative orientation of the other modules versus $G_{core}$ and $M_{core}$ was different (Supplementary Fig. 7c, d). In particular, the distance between the $M_{lid}$ and N domain of *Te*CphA1 is shorter than that of *Su*CphA1 (Supplementary Fig. 7d).

### Functional roles of $G_{core}$ and $G_\omega$ in adding aspartate to cyanophycin primer peptide
The *Te*CphA1 structure in the substrate-bound state visualizes the initial state of the catalytic reaction for the ATP-dependent addition of aspartate to the C-terminal backbone carboxy group of cyanophycin (Fig. 1b and Supplementary Fig. 1). In the cryo-EM map processed with $C_1$ symmetry, all the *Te*CphA1 protomers in the tetramer bound a cyanophycin primer peptide, (β-Asp-Arg)$_4$, on the surface of $G_{core}$ (Fig. 1a). The cryo-EM map showed a clear density for the C-terminal β-Asp-Arg dipeptide unit (4th unit) and the other Asp backbones with a hook-like shape (Fig. 2a). The Arg sidechain of the 3rd unit partially showed weak density. Since the Arg sidechains of the 1st and 2nd units

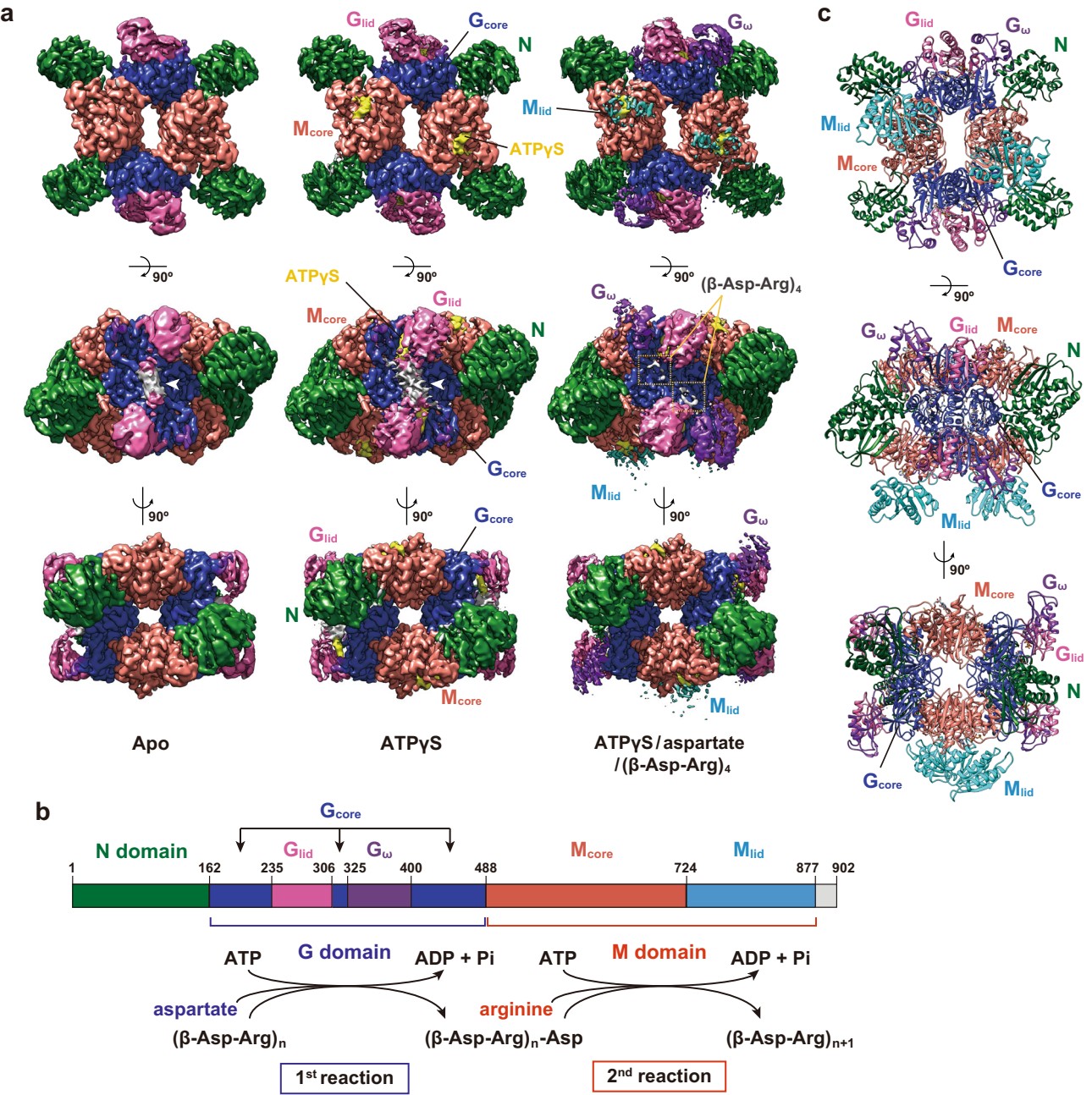

**Fig. 1 | Tetrameric structure and domain architecture of *Te*CphA1. a** Cryo-EM maps of *Te*CphA1 in the apo, ATPγS-bound, and substrate (ATPγS/aspartate/(β-Asp-Arg)₄)-bound states. The maps are divided into six modules (N, G_{core}, G_{lid}, G_{ω}, M_{core}, and M_{lid}) and ATPγS, which are shown in different color coordination. The map of (β-Asp-Arg)₄ is highlighted with a yellow dashed-line box in the density map of substrate-bound *Te*CphA1. A lump on the extra map (arrowhead) is observed between two G_{lid} modules in the apo and ATPγS-bound states. Each map is viewed from a twofold axis of tetrameric *Te*CphA1. **b** Residue range of each module on the *Te*CphA sequence and the reaction scheme of the G and M domains. The color coordination of each module is consistent with that in Panel **a**. **c** Ribbon diagram of the tetrameric *Te*CphA structure in the substrate-bound state. Each protomer is divided into the above-defined modules with the same color coordination as Panel **a**.

were invisible in the cryo-EM map, these models could not be built in the substrate-bound *Te*CphA1 structure. In contrast, the Arg sidechain of the 2nd unit was visible in the cryo-EM map of *Su*CphA1[13], and its guanidium group was located near Ala188. This residue is substituted with Phe on many cyanobacterial CphA1, including *Te*CphA1 (Fig. 2a and Supplementary Fig. 9). The residue type at position 188 may affect the Arg-sidechain orientation of the 2nd unit on the active site of G_{core}. In the *Te*CphA1 structure, the C-terminal backbone carboxy group and Arg sidechain of the 4th unit interact with Arg309 (the nearest atomic distance, 2.9 Å) and Glu215, which is derived from another protomer forming the G_{core}-G_{core} dimer (Glu215′) (2.8 Å), respectively (Fig. 2a and

Supplementary Fig. 8a). Since the catalytic activity of *Te*CphA1 was impaired by the Ala substitutions of Glu215 and Arg309 (Fig. 2b and Supplementary Fig. 9), the interaction between the G_{core}-G_{core} dimer and the 4th unit seems to be required for the activity of *Te*CphA1.

The mutations of Glu215 and Arg309 (E215A and R309A) of the G domain were commonly analyzed in the two studies on *Te*CphA1 and *Su*CphA1[13]. Of the two, R309A (and its corresponding mutation in *Su*CphA1) showed almost no activities. Both residues seem to be essential for the catalytic activities of *Te*CphA1 and *Su*CphA1. On the other hand, while the assay conditions were not the same between the two, the E215A mutation was different between *Te*CphA1 and *Su*CphA1.

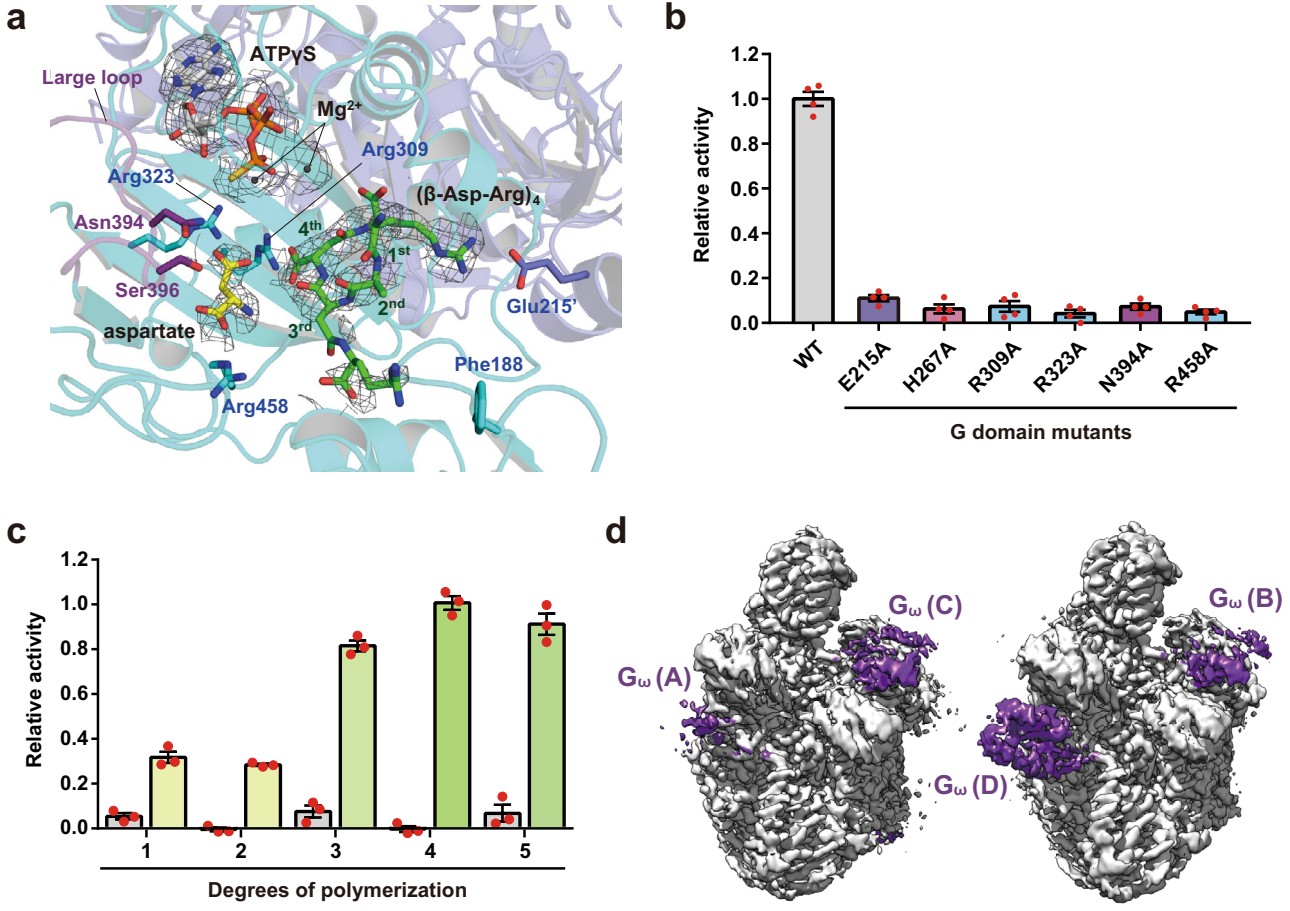

**Fig. 2 | Aspartate and cyanophycin primer peptide recognition at the G domain of *Te*CphA1. a** Binding site of ATPγS, the substrate aspartate, and a cyanophycin primer peptide, (β-Asp-Arg)$_4$, on the surface of the G domain. The cryo-EM map carved 2 Å around Mg$^{2+}$, ATPγS, the substrate aspartate, and (β-Asp-Arg)$_4$ at contour level 5. Individual chains of the G domain dimer are colored cyan (chain D) and medium blue (chain B). The purple structure shows the large loop of G$_\omega$ of chain D. The repeating units of (β-Asp-Arg)$_4$ are numbered from the N-terminus. **b** Relative enzymatic activity of *Te*CphA1 (WT) and each G domain mutant, which was measured by detecting released phosphates. Data are presented as mean ± standard error of the mean (SEM) (*n* = 4 independent experiments). **c** Relative activity of *Te*CphA1 using the primer β-Asp-Arg peptide with different degrees of polymerization (DP = 1–5). The data of the reaction solution with (right) and without (left) *Te*CphA1 are presented as mean ± SEM (*n* = 3 independent experiments). **d** Cryo-EM map of substrate-bound *Te*CphA1. Densities of the four G$_\omega$ modules of the *Te*CphA1 tetramer are shown in purple.

The E215A of *Te*CphA1 lost most of the catalytic activity (Fig. 2b), but the corresponding mutation of *Su*CphA1 retained nearly half the activity. Since the interaction between E215 and the C-terminal Arg sidechain of cyanophycin contributes more to the cyanophycin binding in *Te*CphA1 than in *Su*CphA1, the results of this mutational analysis seem to be explained by the difference in the structures. Although the importance of the Arg sidechains for the lack of poly-Asp polymerase activity has been explained by the structural observation of *Su*CphA1[13,18], our results showed that the Arg sidechain of the 4th unit is a major contributor to (β-Asp-Arg)$_4$ recognition and determines the position of the 4th Asp backbone at the active site of the G domain to avoid polymerization progression with the aspartate backbone alone. Since (β-Asp-Arg)$_4$ is well accommodated at the binding site of G$_{core}$, a cyanophycin molecule with at least 3–4 dipeptides long seems to be suitable as a primer peptide for *Te*CphA1 activity (Fig. 2c). This observation is consistent with the previous results of the other CphA1[18]. According to the common structural features between *Te*CphA1 and *Su*CphA1, the cyanophycin primer peptide seems to adopt a hook-like shape at the active site of CphA1 (Fig. 2a).

While (β-Asp-Arg)$_4$ was observed in all protomers, the substrate aspartate was observed only in one protomer (chain D) (Fig. 2a). The map intensity of G$_\omega$ was significantly different among the four protomers, and G$_\omega$ of chain D showed the highest map intensity of the four

G$_\omega$ (Fig. 2d). The substrate aspartate was visualized in the most stable G$_\omega$. The other G$_\omega$ modules seemed to be more mobile than those in chain D. *Te*CphA1 required four substrates, L-aspartate, L-arginine, (β-Asp-Arg)$_4$, and ATP, for the catalytic reaction (Supplementary Fig. 10a). Steady-state kinetic analysis showed a typical Michaelis-Menten kinetics for L-arginine, (β-Asp-Arg)$_4$, and ATP (Supplementary Fig. 10b). While we have determined the apparent $K_m$ value for ATP, *Te*CphA1 has two active sites in the G and M domains for the different catalytic reactions with ATP. As observed for the other CphA1 enzyme[19], the sequential condensation reaction of *Te*CphA1 may be rate-limited primarily by the site with lower affinity for ATP when the $K_m$ values for ATP differ between the G and M domains. Unlike these substrates, aspartate acts as a positive effector of the catalytic reaction with the Hill coefficient ($h$) 1.92 ± 0.07. In addition, $K_{half}$ and $K_{prime}$ (= $K_{half}^h$) were estimated as 18.0 ± 1.1 mM and 254 ± 21 mM, respectively. As shown in Supplementary Fig. 10b, the catalytic activity of *Te*CphA1 increases with the increasing aspartate concentration; the cellular aspartate concentration may be a key regulator for the catalytic activity of *Te*CphA1. The results of the kinetic analysis for aspartate (Supplementary Fig. 10b) suggest that 100 mM aspartate is appropriate for the cryo-EM analysis. While it is unclear why the aspartate molecule was visible in only one subunit, the cryo-EM structure of *Te*CphA1 with ATPγS, (β-Asp-Arg)$_4$, and aspartate may be a

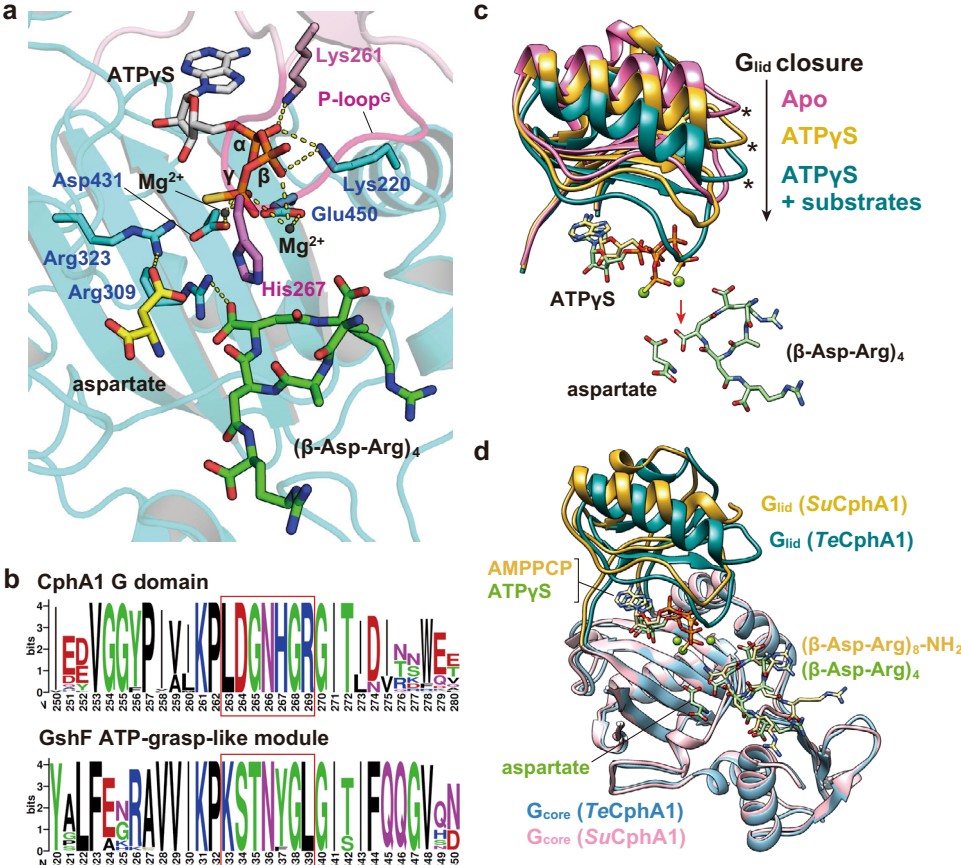

**Fig. 3 | G_lid movement dependent on the binding of ATP and substrates, aspartate and (β-Asp-Arg)₄.** **a** Residue arrangement around ATPγS, the substrate aspartate, and the C-terminal backbone carboxy group of (β-Asp-Arg)₄. Hydrogen bonds and Mg²⁺ coordination are shown as yellow dashed lines. **b** Sequence logo depicting different conserved motifs between the P-loop^G of CphA1 and the P-loop of the GshF ATP-grasp-like module. The P-loop sequences are shown in red boxes. The numbers under the logo correspond to the residue numbers of *Te*CphA1 and

GshF from *Pasteurella multocida*. **c** Structural comparison of G_lid among three states of *Te*CphA1. A comparison was performed after superposing the G_core modules of the three states. Green spheres represent two Mg²⁺ ions. Asterisks and red arrows indicate the position of Lys293 and the C-terminal backbone carboxy group of (β-Asp-Arg)₄, respectively. **d** Superposed structures using the G_core modules of the substrate-bound *Te*CphA1 and the AMPPCP/(β-Asp-Arg)₈-NH₂-bound *Su*CphA1 structure (PDB 7LGJ)[13].

snapshot of the equilibrium state of complex formation between the enzyme and aspartate.

The substrate aspartate was accommodated in a small space beside Ser396 (Fig. 2a). This binding mode seemed to be conserved in *Su*CphA1 because *Su*CphA1 activity was lost by the S396W mutation[13]. The substituted Trp residue prevented the reaction of Asp backbone elongation by occupying the space for substrate aspartate binding (Fig. 2a). Residues contributing to substrate aspartate recognition are conserved in the cyanobacterial CphA1 enzymes (Fig. 2a and Supplementary Fig. 9). Asn394, a residue on the large loop of G_ω (residues 389–399), and Arg323 interacted with the β-carboxy group of the substrate aspartate at distances of 2.5 Å and 3.4 Å, respectively (Fig. 2a). The α-carboxy group of the substrate aspartate was close to the sidechain of Arg458 (2.5 Å) in the *Te*CphA1 structure. Each Ala mutation of Asn394, Arg323 or Arg458 impaired *Te*CphA1 activity (Fig. 2b). These observations are consistent with the notion that G_ω functions as a module for substrate aspartate recognition.

**Catalytically important residues in G_core and G_lid for Asp backbone elongation**

The G domain adopts an ATP-grasp fold that requires nucleophilic attack of the C-terminal backbone carboxy group of cyanophycin on ATP for Asp backbone elongation[13,20]. However, since these two functional groups are negatively charged, it is necessary to avoid

electrostatic repulsion between them. The ATP bound to the ATP-grasp fold typically interacts with one to three Mg²⁺ ions[20]. In the G domain of *Te*CphA1, the γ-phosphate chelates two Mg²⁺ ions together with two acidic residues, Asp431 and Glu450 (Figs. 2a and 3a). This structural feature is similar to *Su*CphA1[13] and other ATP-grasp enzymes, such as bacterial glutathione synthetases[21] and N⁵-carboxyaminoimidazole ribonucleotide synthetase (PurK)[20]. In addition, the α- and β-phosphates of ATPγS interact with the positively charged sidechains of Lys261 and Lys220, respectively (Fig. 3a). While the triphosphate of ATP is neutralized overall by these interactions, there are still two conserved charged residues, Arg309 and Arg323, around the γ-phosphate. The substrate-bound *Te*CphA1 structure shows that these residues electrostatically interact with the C-terminal backbone carboxy group of (β-Asp-Arg)₄ and β-carboxy group of the bound aspartate (Fig. 3a and Supplementary Fig. 9).

The substrate-bound-state structure suggests that the phosphate-binding loop of G_lid (hereafter P-loop^G, residues 263–269) (Fig. 3b) plays a critical role in the catalytic reaction. In the substrate-bound state, His267 on the P-loop^G is located between γ-phosphate and the C-terminal backbone carboxy group of (β-Asp-Arg)₄ (Fig. 3a). This His267 position seems to be suitable to stabilize an acylphosphate intermediate together with Arg309, following the nucleophilic attack of the substrate aspartate. The importance of these residues was confirmed by the conservation of cyanobacterial CphA1 enzymes, including *Su*CphA1 and decreased activity of a mutant substituted with

Ala both in $Te$CphA1 and $Su$CphA1 (Figs. 2b and 3b)[13]. This loop corresponds to the B loop of PurT and PurK[22], which are ATP-grasp enzymes of *E. coli*. These enzymes function in the de novo purine biosynthetic pathway, and the B loop is required to efficiently generate an acylphosphate intermediate in these enzymes. However, His267, which seems to be important to stabilize the acylphosphate intermediate in $Te$CphA1, is not conserved in PurT and PurK[22].

### $G_{lid}$ motion in the catalytic mechanism of Asp backbone elongation

Due to the charge neutralization of the γ-phosphate of ATP and the C-terminal backbone carboxy group of cyanophycin, these two groups can approach each other until reaching the distance of nucleophilic attack. A comparison of the three states of the $Te$CphA1 structure (Fig. 3c) revealed conformational changes of $G_{lid}$ induced by ATP, the substrate aspartate and (β-Asp-Arg)$_4$ binding. These conformational changes seem to contribute to optimizing the relative dispositions of ATP, aspartate, and (β-Asp-Arg)$_4$ for the catalytic reaction. The binding of ATPγS causes $G_{lid}$ to move toward $G_{core}$ by 4.6 Å, which is the distance between the $C_\alpha$ atoms of Lys293 (Fig. 3c, asterisk), and the substrate aspartate and (β-Asp-Arg)$_4$ binding induces a further shift of $G_{lid}$ by 4.3 Å. This $G_{lid}$ movement seems to change the binding position of ATPγS toward (β-Asp-Arg)$_4$. Interestingly, $G_{lid}$ and ATPγS in the substrate-bound $Te$CphA1 structure are located at a position closer to $G_{core}$ than those in the $Su$CphA1 structure bound with the cyanophycin analog (β-Asp-Arg)$_8$-NH$_2$ (Fig. 3d). Since the residues contributing to the reaction of Asp backbone elongation in the G domain are conserved between $Te$CphA1 and $Su$CphA1 (Supplementary Fig. 9), the binding of the substrate aspartate seems to modulate the orientation of ATPγS together with $G_{lid}$.

While the binding of ATP, the substrate aspartate, and (β-Asp-Arg)$_4$ seems to cause the rearrangement of their relative orientations for the catalytic reaction, the γ-phosphate of ATP is less susceptible to nucleophilic attack by the C-terminal backbone carboxy group of (β-Asp-Arg)$_4$, because they are located at a distance of ~6 Å (Fig. 3c). Although $G_{lid}$ moves toward the active site to change the relative positions of ATP and (β-Asp-Arg)$_4$, a further mechanism is required to bridge this distance gap for the catalytic reaction to proceed. One of the possible mechanisms is the thermal motion of $G_{lid}$, which is mentioned in relation to $Su$CphA1[13]. Since the cryo-EM structures of $Te$CphA1 only represent the average position in the equilibrium of the $G_{lid}$ motion, it may be possible to consider that the protein dynamics allow these two molecules to occasionally access the distance of the catalytic reaction.

While the relative orientation of ATPγS and (β-Asp-Arg)$_4$ in the substrate-bound state needs a greater change for the catalytic reaction, the relative orientation of (β-Asp-Arg)$_4$ and the substrate aspartate in the G domain seems to be suitable for ATP-dependent backbone elongation (Fig. 1b and Supplementary Fig. 1). The amino group of substrate aspartate is located 4.1 Å from the carboxy group at the C-terminus of (β-Asp-Arg)$_4$, and the position of the carboxy group may be adjusted by phosphorylation (Figs. 2a and 3c).

### Structural and functional roles of $M_{lid}$ in Arg sidechain condensation

In the substrate-bound state, $M_{core}$ was clearly visible in the four protomers of the $Te$CphA1 tetramer. ATPγS interacts with the conserved P-loop of $M_{core}$ (hereafter P-loop$^M$, residues 495–501) (Fig. 4a and Supplementary Fig. 9). Lys499 and an Mg$^{2+}$ ion are located between the β-phosphate and γ-thiophosphate groups of ATPγS, and the Mg$^{2+}$ ion is chelated by Thr500, Thr522, and Glu558. These interactions with ATP are highly conserved in the Mur ligases, which synthesize the peptide stem of bacterial peptidoglycan[23,24]. Since the Ala mutation of Lys499 impaired $Te$CphA1 activity (Fig. 4b), the interaction between Lys499 and ATP is also critical to the catalytic activity of CphA1 enzymes.

While ATPγS is visible in the $M_{core}$ modules of all protomers, the cryo-EM densities for $M_{lid}$ were observed weakly only in protomers A and B (Supplementary Fig. 11), suggesting a relatively mobile nature of $M_{lid}$, even in the substrate-bound state. In protomers A and B, $M_{lid}$ and $M_{core}$ sandwich ATPγS (Figs. 1a and 4a), and Arg731 and His748 in $M_{lid}$ interact with the α-phosphate and γ-thiophosphate groups of ATPγS. Since $M_{lid}$ was not observed in the ATPγS-bound state, $M_{lid}$ is unlikely essential to the stable binding of ATPγS.

However, since the deletion mutant of $M_{lid}$ ($\Delta M_{lid}$) showed substantially reduced activity of $Te$CphA1, the $M_{lid}$ module is necessary for the efficient catalytic reaction of $Te$CphA1 (Fig. 4b) and seems to play a significant role in ATP-dependent Arg sidechain condensation. Superposition of $Te$CphA1 in apo- and ATPγS-bound structures reveals a conformational change of Phe692 in $M_{core}$ upon ATPγS binding, and Phe692 and Asn497 sandwich the adenine ring of ATPγS (Fig. 4c). This sidechain orientation of Phe692 causes steric hindrance with the methyl group of Ala755 in $M_{lid}$ (Fig. 4c and Supplementary Fig. 12). On the other hand, the sidechain position of Phe692 in the apo form inhibits ATP binding but not the interaction between $M_{core}$ and $M_{lid}$ (Fig. 4c). The flexible nature of $M_{lid}$ and the conformational change of Phe692 seem to cause mutually exclusive binding conformations of ATP and $M_{lid}$, which may facilitate the exchange of ATP and ADP in the reaction cycle. The invisible $M_{lid}$ in protomers C and D can be explained by mutually exclusive binding conformations, and protomers C and D are considered ATPγS-binding states. Intriguingly, protomers A and B of the substrate-bound state showed simultaneous binding of ATPγS and $M_{lid}$ to $M_{core}$. We consider that $M_{lid}$ and ATPγS in protomers A and B are in a metastable state that can be realized due to a high concentration of ATPγS. The interaction between Ala755 and Phe692 in protomers A and B may be energetically unfavored and thus easily changed to the $M_{lid}$-binding or ATPγS-binding state under lower ATPγS concentrations.

While ATPγS was visible on all M domains, (β-Asp-Arg)$_4$ was not observed on the M domain of the substrate-bound $Te$CphA1 structure. This observation is consistent with the previous result that the Arg-unmodified Asp residue at the C-terminus of cyanophycin is required to bind to the M domain in the $Su$CphA1 structure[13]. When the $Te$CphA1 structure is superposed with the $Su$CphA1 structure in complex with the cyanophycin analog (β-Asp-Arg)$_8$-Asn[13], the cyanophycin analog is located between the N domain and $M_{core}$ of $Te$CphA1. Interestingly, we found a densely charged region with multiple Arg and Asp/Glu residues on the surface of $M_{lid}$ (Fig. 4d). These residues seem to interact with the cyanophycin analog, as observed in $Su$CphA1. $M_{lid}$ may contribute to guiding cyanophycin into the active site of $M_{core}$. In the Mur ligases, the interaction with the substrate induces a rotation of the C-terminal domain, which corresponds to $M_{lid}$ of CphA1 enzymes, resulting in a closed conformation[15,17]. However, the orientation of $M_{lid}$ is almost unchanged by the substrate (cyanophycin) interaction in $Su$CphA1 (RMSD = 0.158 Å, 2,583 atoms of $M_{core}$ and $M_{lid}$)[13].

## Discussion

Cyanophycin is a unique biopolymer that consists of the β-Asp-Arg dipeptide as a repeating unit (Supplementary Fig. 1) and is produced by CphA1-dependent peptide synthesis. Our understanding of this polymerization process has been greatly improved by a recent report that provided structural snapshots of CphA1[13]. However, the catalytic and regulatory mechanisms of this reaction have not been fully elucidated. In the present study, we determined the CphA1 structure with a complete set of substrates for the reaction of the G domain, aspartate and (β-Asp-Arg)$_4$, which visualizes the initial state for ATP-dependent addition of aspartate to the C-terminal backbone carboxy group of a cyanophycin primer peptide. This structure proposes a feasible catalytic mechanism for the Asp backbone elongation reaction that occurs in the G domain (Fig. 5). In the apo state, $G_{lid}$ adopts an open conformation and is the furthest away from $G_{core}$. ATP binding induces the movement of $G_{lid}$ toward $G_{core}$ to sandwich an ATP molecule, whereas

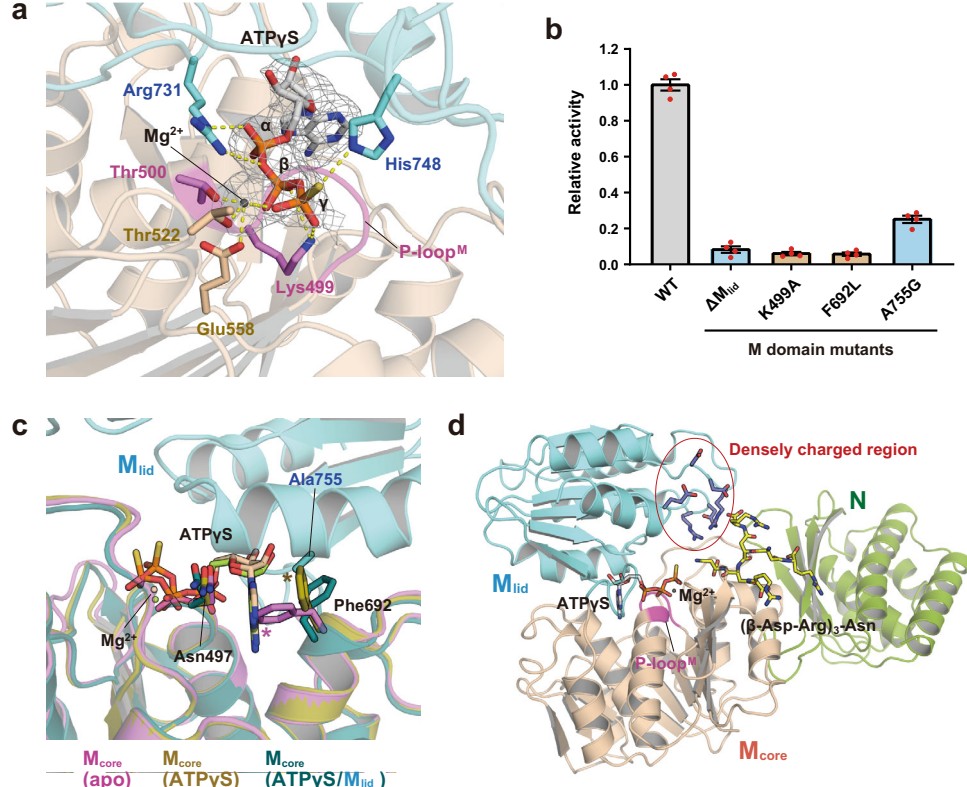

**Fig. 4 | Structural bases of $M_{lid}$ for ATP-dependent Arg sidechain condensation to cyanophycin. a** Binding site of ATPγS between $M_{core}$ (wheat) and $M_{lid}$ (cyan) on the surface of the M domain. The cryo-EM map carved 2 Å around $Mg^{2+}$ and ATPγS at contour level 10. The magenta structure shows the P-loop$^M$. Hydrogen bonds and $Mg^{2+}$ coordination are shown as yellow dashed lines. **b** Relative enzymatic activity of $Te$CphA1 (WT) and each M domain mutant, including the deletion mutant of $M_{lid}$ (Δ$M_{lid}$). Enzyme activities were measured by detecting the released phosphates. Data are presented as mean ± standard error of the mean (SEM) ($n = 4$ independent experiments). **c** Superposed structures of the M domain in three $Te$CphA1 states. Asterisks (*) represent the steric crashes of Phe692 with the adenine ring of ATPγS (pink) and Ala755 (ocher). **d** Localization of a cyanophycin analog, (β-Asp-Arg)$_3$-Asn, on the N and M domains of $Te$CphA1. The model of (β-Asp-Arg)$_3$-Asn was created by the superposition between the $M_{core}$ modules of the substrate-bound $Te$CphA1 structure and the ATP/(β-Asp-Arg)$_8$-Asn-bound $Su$CphA1 structure (PDB 7LGQ)[13].

$G_\omega$ remains flexible. Cyanophycin and aspartate binding further changes $G_{lid}$ to a closed conformation, and the orientation of $G_\omega$ is fixed to localize aspartate in a suitable position for the reaction by the interaction of Asn394 on the large loop with the substrate. Thus, ATP, the substrate aspartate, cyanophycin, and residues, including His267 on the P-loop$^G$, are arranged to initiate the reaction of Asp backbone elongation. The C-terminal backbone carboxy group of cyanophycin is activated by ATP to form a high-energy acylphosphate intermediate required for the ATP-grasp enzymes[22], which is stabilized by His267 on the P-loop$^G$ together with Arg309. The amino group of the substrate aspartate nucleophilically targets the carboxy carbon of the acylphosphate, thus generating a tetrahedral intermediate in which an oxyanion is stabilized by the interaction with Arg309 and Gly456. A new peptide bond forms between cyanophycin and the substrate aspartate with the release of phosphate.

The G domain of CphA1 enzymes adopts a fold similar to the ATP-grasp-like module of bacterial hybrid-type glutathione synthetase, GshF, which catalyzes peptide bond formation between the amino group of glycine and the C-terminal backbone carboxy group of γ-glutamyl cysteine[13,25]. The amino acid sequence of the P-loop$^G$ in CphA1 is different from that in GshF. In particular, the P-loop$^G$ has a His residue (His267) in CphA1 but a Tyr or Phe residue in GshF (Fig. 3b). In addition, Glu215 and Arg458 of $Te$CphA1 are substituted with Leu and Met/Tyr in GshF, respectively. These different residues of CphA1 from GshF seem to be optimized for the catalytic reaction for the Asp backbone elongation of cyanophycin. Mutational analysis suggested that His267 plays a critical role in the catalytic reaction for $Te$CphA1

(Figs. 2b and 5). Glu215 and Arg458 are required for the recognition of (β-Asp-Arg)$_4$ and the substrate aspartate at the active site of $G_{core}$, respectively (Fig. 2a, b).

Although Tlr2120 from *Thermosynechococcus elongatus* BP-1 has been found to be the only CphA1 enzyme that synthesizes cyanophycin in a primer-independent manner[26], the other known CphA1 enzymes show primer-dependent activity. Efficient CphA1 activity requires a cyanophycin primer at least 3–4 dipeptides long (Fig. 2c)[18]. Hence, to rapidly initiate cyanophycin synthesis upon the transition to the $N_2$ fixation phase, cyanophycin primer peptides must be available in cyanobacterial cells. Another critical factor for CphA1 activity is the cellular level of aspartate. In the process of $N_2$ fixation, aspartate is produced by the incorporation of nitrogen into oxaloacetate and is consumed directly for cyanophycin synthesis[9]. Aspartate is also required as a nitrogen donor for arginine biosynthesis in cyanobacteria[27]. Therefore, the intracellular aspartate concentration is one of the factors most likely to affect the progression of cyanophycin synthesis. The steady-state kinetics of $Te$CphA1 suggests an allosteric effect with positive cooperativity toward aspartate, and the reaction is, in fact, accelerated with an increasing aspartate concentration up to quite a high concentration range (Supplementary Fig. 10b). The structural comparison between the substrate-bound $Te$CphA1 and the $Su$CphA1 bound with the cyanophycin analog (β-Asp-Arg)$_8$-NH$_2$ showed that the binding of aspartate modulates the orientation of ATP together with $G_{lid}$ (Fig. 3d). This aspartate-mediated regulation of $G_{lid}$ seems reasonable to control the CphA1 activity associated with storage of fixed nitrogen in cyanobacteria to avoid wasteful phosphorylation

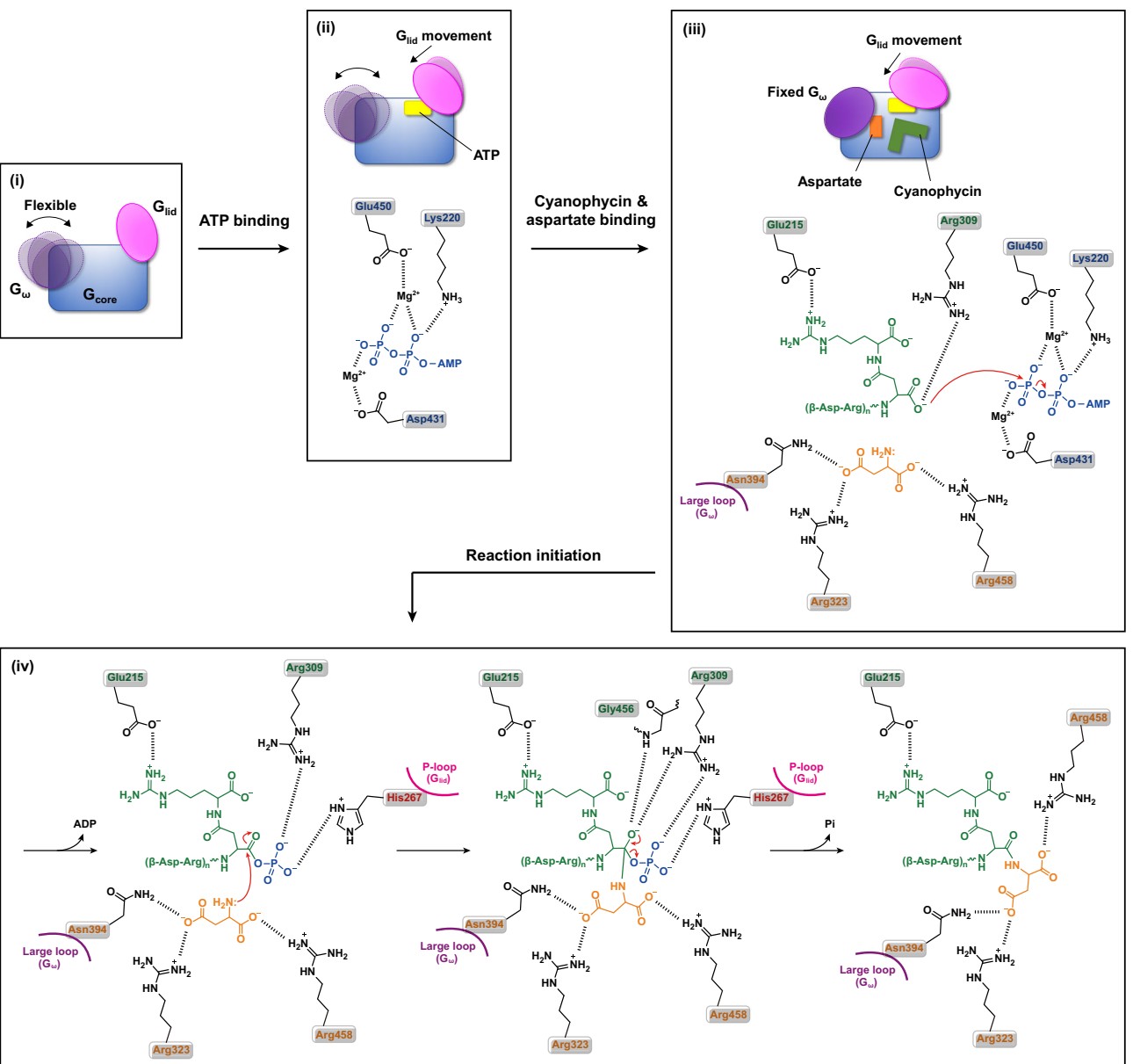

**Fig. 5 | Schematic diagram of the catalytic reaction for aspartate addition to the C-terminal backbone carboxy group of cyanophycin.** (i) G domain from a protomer in the apo state. Three modules, $G_{core}$, $G_{lid}$, and $G_\omega$, are shown in blue, pink, and purple, respectively. (ii) ATP-bound state. The chemical structure of ATP is depicted as AMP diphosphate. (iii) Substrate-bound state. The orientation of $G_{lid}$ further changes compared to the ATP-bound state. (iv) Feasible catalytic

mechanism in the G domain. In this mechanism, a tetrahedral intermediate is formed by nucleophilic attack to an acylphosphate intermediate of cyanophycin by the amino group of aspartate. Arg309 is assumed to stabilize the acylphosphate intermediate and the tetrahedral intermediate together with His267 and Gly456, respectively.

of primer peptides under conditions in which aspartate is not available to CphA1 as a substrate.

In the apo and ATPγS-bound states, a large density was observed between two $G_{lid}$ modules and over the binding site of (β-Asp-Arg)$_4$ (Fig. 1a, arrowhead). CphA1 is known to synthesize cyanophycin in *E. coli* cells[28]; hence, cyanophycin is the most likely candidate for the molecule with increased density. In addition, the polymerization reaction of CphA1 often terminates up to a polymer length of 25–30 kDa[26]. The mobility of $G_{lid}$ is likely to be affected by the association of a high-molecular-weight molecule into the space between two $G_{lid}$ modules, and the regulation of $G_{lid}$ motion may also be related to the length determination and polymerization termination of cyanophycin, although this mechanism is a question that needs to be addressed in

further study. On the other hand, the windshield wiper model using the N domain has been proposed as a promising model for predicting the efficiency of reactions using two distant active sites alternately[13]. CphA2, which synthesizes cyanophycin by directly linking β-Asp-Arg dipeptides, lacks not only the P-loop$^M$ but also the $M_{lid}$ module to abolish the activity for Arg sidechain condensation[29], which is supported by our mutational data of *Te*CphA1, which does not exert the CphA2 activity (Fig. 4b and Supplementary Fig. 10a). This molecular evolution may imply that the presence of $M_{lid}$ is unfavorable for the reaction to proceed efficiently only in the G domain, rather than alternately in the two domains. The mobility of $M_{lid}$ and its densely charged region (Fig. 4c) may play a role in the reactivity of both active sites together with the N domain.

## Methods

### Materials

Cyanophycin primer peptides β-Asp-Arg, (β-Asp-Arg)$_2$, (β-Asp-Arg)$_3$, (β-Asp-Arg)$_4$, and (β-Asp-Arg)$_5$ were synthesized via the solid phase method by Sangon Biotech (Shanghai, China). The synthesized peptides were confirmed using a TripleTOF 5600+ system (Sciex) with a TurboIonSpray (TIS) probe (Turbo V ion source, Sciex). Time of flight-mass spectroscopy (TOF-MS) spectra (m/z range, 100–2000 Da) were recorded in positive-ion mode with the following parameters: curtain gas, 15 psi; ion source gas 1, 15 psi; ion source gas 2, 0 psi; ion spray floating voltage, 5500 V; and interface heater temperature, 400 °C. The mass spectrometer was recalibrated using an APCI positive calibration solution (Sciex). The obtained spectra are listed in Supplementary Fig. 13. All the other chemicals were purchased from Sigma–Aldrich or FUJIFILM Wako Pure Chemical Corporation.

### Expression and purification of *Te*CphA1

The gene encoding *Te*CphA1 (ENA code: ABG51217) was amplified from the genomic DNA of *T. erythraeum* IMS101 with forward and reverse primers (CphA-for: TATACATATGAAAATCCTCAAACTCCAGA CAT-TACG; CphA-rev: GGTGCTCGAGAATTGAACTTTTTAAAACTT) containing 5′ 10-bp overhangs complementary to the vector sequence for seamless cloning (TransGen Biotech). The target gene was cloned into the pET-22b(+) vector (Novagen) between the restriction sites NdeI and XhoI along with a C-terminal hexa-histidine tag. All *Te*CphA1 mutants were prepared by inverse PCR for site-specific mutagenesis using PrimeSTAR Max DNA polymerase (Takara Bio) and primers listed in Supplementary Table 2.

Recombinant proteins were expressed in the *E. coli* strain BL21(DE3) (Novagen). Cells were inoculated in LB media supplemented with 50 mg ml$^{-1}$ ampicillin and cultured overnight. Upon reaching an optical density of ~0.6 at 600 nm, the temperature was decreased from 37°C to 16°C, and the culture was supplemented with 0.5 mM isopropyl β-D-1-thiogalactopyranoside. Protein expression was performed for 16 h, after which cells were harvested by centrifugation at 5,180 × *g* for 10 min, followed by resuspension in lysis buffer (20 mM Tris-HCl, pH 8.0, 500 mM NaCl, 1 mM dithiothreitol (DTT), and 5 mM imidazole). Cells were lysed by sonication, and cell debris was then removed by centrifugation at 40,000 × *g* for 30 min. The supernatant fraction was loaded onto an equilibrated Ni-nitrilotriacetic acid (NTA) resin (Qiagen). After extensive washing with wash buffer (20 mM Tris-HCl, pH 8.0, 500 mM NaCl, 1 mM DTT, and 20 mM imidazole), the recombinant proteins were eluted with elution buffer (20 mM Tris-HCl, pH 8.0, 500 mM NaCl, 1 mM DTT, and 200 mM imidazole). The eluate was further purified by loading onto a Mono Q column (Cytiva) for anion exchange chromatography, and elution was then performed with a linear gradient of 0 to 1.0 M NaCl. After concentration with Vivaspin 20 devices (30 kDa cutoff; Sartorius), the protein solution was loaded onto a Superdex200 10/300 GL column (Cytiva) for size exclusion chromatography with buffer containing 20 mM Tris-HCl (pH 8.0), 400 mM NaCl, and 1 mM DTT. The purified *Te*CphA1 and its mutants were concentrated with Vivaspin 20 devices and then stored in buffer for size exclusion chromatography at –80°C until use.

### Grid preparation and cryo-EM data collection

Cryo-EM samples were prepared by mixing 24 μM *Te*CphA1 with each solution. The final sample compositions were as follows: (1) apo state, 12 μM *Te*CphA1 in a solution containing 20 mM Tris-HCl (pH 8.0), 400 mM NaCl, and 1 mM DTT; (2) ATPγS-bound state, 2 μM *Te*CphA1 in a solution containing 50 mM Tris-HCl (pH 8.2), 200 mM NaCl, 20 mM KCl, 20 mM MgCl$_2$, 1 mM DTT, 4 mM ATPγS, 5 mM L-aspartate, 0.5 mM L-arginine, and 10 mM (β-Asp-Arg)$_4$; and (3) substrate-bound

state, 4 μM *Te*CphA1 in a solution containing 50 mM Tris-HCl (pH 8.2), 67 mM NaCl, 20 mM KCl, 20 mM MgCl$_2$, 1 mM DTT, 10 mM ATPγS, 100 mM L-aspartate, and 20 mM (β-Asp-Arg)$_4$. Three microliters of sample was applied to a holey carbon grid (Quantifoil, Cu, R1.2/1.3, 300 mesh) rendered hydrophilic by a 30 s glow-discharge in air (11 mA current with PIB-10). The grid was blotted for 15 s with a blot force of 0 (sample 1), for 15 s with a blot force of 10 (sample 2), and for 10 s with a blot force of 10 (sample 3) and flash-frozen in liquid ethane using Vitrobot Mark IV (Thermo Fisher Scientific) at 18 °C and 100% humidity.

For apo and ATPγS-bound states, cryo-EM datasets were acquired with a Talos Arctica (Thermo Fisher Scientific) transmission electron microscope (TEM) operating at 200 kV in nanoprobe mode using EPU software for automated data collection (Supplementary Fig. 5a). The movie frames were collected by a 4 k × 4 k Falcon 3 direct electron detector (DED) in electron counting mode at a nominal magnification of 120,000, which yielded a pixel size of 0.88 Å pixel$^{-1}$. Forty-nine and fifty movie frames were recorded at an exposure of 1.02 and 1.00 electrons per Å$^2$ per frame, corresponding to a total exposure of 50 e Å$^{-2}$, respectively. The defocus range was –1.0 to –2.5 μm for the apo and ATPγS-bound states. The dataset for the substrate-bound state was obtained with a 300 kV Titan Krios G3i TEM (Thermo Fisher Scientific) equipped with a Gatan Imaging Filter (GIF), Quantum energy filter (Gatan), and a K3 Summit DED (Gatan) (Supplementary Fig. 5a). The dataset was acquired in counting mode with a nominal magnification of 105,000 (yielding a calibrated pixel size of 0.83 Å pixel$^{-1}$) using SerialEM software[30]. Forty-nine movie frames were recorded at an exposure of 1.00 electrons per Å$^2$ per frame, corresponding to a total exposure of 49 e Å$^{-2}$ and –0.84 to –1.8 μm of defocus range. Data collection details are listed in Supplementary Table 1.

### Cryo-EM data processing

For the apo state dataset, 2,109 acquired movies were subjected to beam-induced motion correction by MotionCor2[31]. The contrast transfer function (CTF) parameters were calculated using Gctf[32]. A total of 340,000 particles were fully automatically selected using SPHIRE crYOLO[33,34] with a generalized model using a selection threshold of 0.1. The subsequent processes were performed using RELION3.1[35]. The particles were extracted by downsampling to a pixel size of 2.64 Å pix$^{-1}$ and subjected to 2D classification. A total of 180,018 particles (17 classes) that displayed secondary-structural elements were selected for ab initio map reconstruction. $D_2$ symmetry was imposed on the generated ab initio map and used as an initial reference map for the 3D classification assuming asymmetry ($C_1$). A total of 103,041 particles associated with the best 3D class with the highest resolution were recentered, re-extracted with a pixel size of 0.88 Å pix$^{-1}$ and subjected to 3D refinement with $D_2$ symmetry. Three rounds of CTF refinement[35] and Bayesian polishing[36] followed by 3D refinement with a soft-edged 3D mask yielded a $D_2$ map with 3.11 Å resolution. To further improve the resolution, additional particles were selected using SPHIRE crYOLO[33,34] with a relaxed selection threshold of 0.02. A total of 556,730 particles were extracted by downsampling to a pixel size of 2.64 Å pix$^{-1}$ and subjected to 2D classification. A total of 429,149 particles (14 classes) that displayed secondary-structural elements were selected. The $D_2$ map with a resolution of 3.11 Å obtained above was used as a reference map for the 3D classification assuming asymmetry ($C_1$). A total of 426,572 particles associated with the best three 3D classes were recentered and re-extracted with a pixel size of 0.88 Å pix$^{-1}$ and subjected to 3D refinement with $D_2$ symmetry. Three rounds of CTF refinement[35] and Bayesian polishing[36] followed by 3D refinement with a soft-edged 3D mask yielded a 3.03 Å resolution map. Then, no-alignment 3D classification was conducted ($D_2$ symmetry, two expected classes, $T = 8$) with a soft-edged 3D mask, and 161,823 particles were selected by

choosing the best 3D class. The last 3D refinement ($D_2$ symmetry, with a mask diameter of 260 Å) with a soft-edged 3D mask and subsequent postprocessing yielded a map with a global resolution of 2.91 Å according to the Fourier shell correlation (FSC) = 0.143 criteria[37]. The details of the cryo-EM data processing are shown in Supplementary Fig. 2, and the associated parameters and statistics are summarized in Supplementary Table 1. Directional FSC plots were calculated using the 3DFSC server to evaluate the evenness of the distribution of particle orientations (Supplementary Fig. 5b, c)[38].

For the dataset of the ATPγS-bound state, 1,730 acquired movies were subjected to beam-induced motion correction by MotionCor2[30]. The CTF parameters were calculated using Gctf[32]. A total of 927,603 particles were picked fully automatically using SPHIRE crYOLO[33,34] with a generalized model using a selection threshold of 0.001. The subsequent processes were performed by using RELION3.1[35]. The particles were extracted by downsampling to a pixel size of 2.64 Å pix$^{-1}$ and subjected to 2D classification. The 424,332 particles (16 classes) that displayed secondary-structural elements were selected for ab initio map reconstruction. $D_2$ symmetry was imposed on the generated ab initio map and used as an initial reference map for the 3D classification assuming asymmetry ($C_1$). A total of 210,971 particles associated with the best 3D class with the highest resolution were recentered and re-extracted with a pixel size of 0.88 Å pix$^{-1}$ and subjected to 3D refinement with $D_2$ symmetry. Two rounds of CTF refinement[35] and Bayesian polishing[36] followed by 3D refinement with a soft-edged 3D mask yielded a $D_2$ map with 2.91 Å resolution. Then, no-alignment 3D classification was conducted ($D_2$ symmetry, two expected classes, $T = 16$) with a soft-edged 3D mask, and 49,842 particles were selected by choosing the best 3D class. The last 3D refinement ($D_2$ symmetry, with a mask diameter of 260 Å) with a soft-edged 3D mask and subsequent postprocessing yielded a map with a global resolution of 2.96 Å according to the FSC = 0.143 criteria[37]. The processing strategy is shown in Supplementary Fig. 3, and the associated parameters and statistics are summarized in Supplementary Table 1. Directional FSC plots were calculated using the 3DFSC server to evaluate the evenness of the distribution of particle orientations (Supplementary Fig. 5b, c)[38].

For the datasets of substrate-bound states, all 3,843 acquired movies were dose-fractionated and subjected to beam-induced motion correction implemented in RELION 3.1[35], and 3,735 motion-corrected micrographs were selected with total motion < 34 Å. The CTF parameters were calculated using CTFFIND4[39], and 3,730 micrographs were then selected with the estimated maximum resolutions <6 Å. A total of 1,371,057 particles were automatically selected from the selected micrographs in RELION 3.1[35] using template images, which were generated from 1,338 particles (9 classes) by 2D classification of the 2,008 manually selected particles. The particles were extracted by downsampling to a pixel size of 2.49 Å pix$^{-1}$ and subjected to several rounds of 2D classification. The 296,927 particles (77 classes) were selected from the last round of 2D classification and applied to two rounds of 3D classification assuming asymmetry ($C_1$). A total of 195,538 particles associated with the best 3D class with the highest resolution of the second round were recentered and re-extracted with a pixel size of 0.83 Å pix$^{-1}$ and subjected to 3D refinement. CTF refinement[35] and Bayesian polishing[36] were applied to the resulting particles to refine the per-particle defocus, beam-tilt, and beam-induced motion corrections. The map generated by the 3D refinement with a soft-edged 3D mask after Bayesian polishing showed a strong indication of $C_2$ symmetry, including the $M_{lid}$ modules. Therefore, $C_2$ symmetry was imposed on the $C_1$ refined map, which was then compared with the original map. The visual inspection showed that the $M_{lid}$ modules were more clearly resolved with the $C_2$ map. Accordingly, $C_2$ symmetry was assumed for the subsequent processing to further improve the resolution. Two rounds of 3D classification were performed using the $C_2$ map, and 358,932 particle images (133 classes) were reselected from

the last round of 2D classification to increase the structural homogeneity of the dataset. A total of 198,918 particles associated with the best class with the highest resolution of the second 3D classification were recentered, re-extracted with a pixel size of 0.83 Å pix$^{-1}$, and then subjected to 3D refinement. After two cycles of CTF refinement[35] and Bayesian polishing[36], the resulting particles were subjected to two rounds of 3D refinement imposing $C_2$ symmetry (first no 3D mask and second with a soft-edged 3D mask). Additionally, to investigate the possible asymmetrical nature of substrate binding as well as the $G_\omega$ and $M_{lid}$ modules of our research interests, the last two rounds of 3D refinement were repeated assuming asymmetry. Postprocessing yielded a map with a global resolution of 2.52 Å for the $C_2$ process and 2.64 for the $C_1$ process according to the FSC = 0.143 criteria[37]. The details of the cryo-EM data processing are shown in Supplementary Fig. 4, and the associated parameters and statistics are summarized in Supplementary Table 1. Directional FSC plots were calculated using the 3DFSC server to evaluate the evenness of the distribution of particle orientations (Supplementary Fig. 5b, c)[38].

## Model building and refinement

Each of the four models of $G_{core}$ and $M_{core}$ in the $Te$CphA1 tetramer was manually built based on the post-processed $D_2$ cryo-EM map of the apo state using Coot[40]. Model refinement was performed using phenix.real_space_refine with the secondary-structure restraints generated by Phenix.secondary_structure_restraints in the PHENIX program suite[41,42]. The built models were fit to the $C_2$ cryo-EM map of the substrate-bound state with UCSF Chimera[43]. The fitted models were readjusted into the map and automatically refined using Coot and PHENIX. The structural models of the other modules, the N domain, $G_{lid}$, $G_\omega$, and $M_{lid}$, were generated using the protein structure modeling tools SWISS-MODEL[44] and AlphaFold2[45] to utilize them as initial models for the further model building to generate relatively weak density maps. SWISS-MODEL was run using a part of the sequence of $Te$CphA1 (residues 218–481) to build the initial models of $G_{lid}$ and $G_\omega$. The model with the highest GMQE (0.75) and QMEAN (–0.84) scores were obtained from the crystal structure of γ-glutamate-cysteine ligase/glutathione synthetase (GshF, PDB 3LN7, sequence identity of 35.5%) as a template. AlphaFold2 predicted the initial models of the N domain (residues 1–161) and $M_{lid}$ (724–879), in which almost the entire region showed high predicted local distance test (pLDDT) scores (>90), except for the terminal residues (residues 160 161, 724–733, and 877–879) and loop regions (residues 58–65 and 826–833). The initial models were fit to the $C_2$ cryo-EM map of the substrate-bound state with UCSF Chimera. Manual model rebuilding and refinement were performed using Coot and PHENIX, respectively. The refined models were applied to the $C_1$ cryo-EM maps of the substrate-bound state and $D_2$ maps of the apo and ATPγS-bound states. After fitting each module to the maps with UCSF Chimera, the models were readjusted to the map and refined using Coot and PHENIX. Finally, ligands were manually fit in the map of the ATPγS-bound and substrate-bound states. The PDB coordinates AGS and 7ID were used for the models of ATPγS and a β-Asp-Arg dipeptide. All the final models were separately validated using MolProbity[46] in PHENIX. The statistics of the model refinement are summarized in Supplementary Table 1. Figures were generated using UCSF Chimera and PyMol (Schrödinger).

## CphA1 activity assays

The reaction mixture contained 2.5–10 μg of purified $Te$CphA1, 50 mM Tris-HCl (pH 8.2), 20 mM KCl, 20 mM MgCl$_2$, 1 mM DTT, 4 mM ATP, 5 mM L-aspartate, 0.5 mM L-arginine, and 2 mM synthetic cyanophycin primer peptide of different lengths (1- to 5-mer). These are essentially the same conditions as in a previous report[19] with a small modification (replacement of 10 mM 2-mercaptoethanol with 1 mM DTT). For enzyme kinetics, the activity of $Te$CphA1 (1 μM as a tetramer) was measured by varying the concentration of one of the four substrates,

ATP, L-aspartate, L-arginine, and (β-Asp-Arg)$_4$. Kinetic analysis revealed that our enzyme-assay conditions for arginine, (β-Asp-Arg)$_4$, and ATP were appropriate. While the concentration of aspartate is rather low for a stable enzyme-activity measurement, we determined the aspartate concentration at 5 mM by referring to an earlier experiment[19]. In the analysis of *Te*CphA1 mutants, a 4-mer peptide, (β-Asp-Arg)$_4$, was used as a primer. The reaction was performed at 30°C for 30 min, with a total reaction volume of 25 μL. To confirm the *Te*CphA1 activity, the amount of released phosphate was measured by the colorimetric molybdenum blue method[47]. The postreaction solution was mixed with 250 μL of molybdate solution (0.5% ammonium molybdate and 0.5 M H$_2$SO$_4$) and 50 μL of SnCl$_2$ solution (2 mg ml$^{-1}$ SnCl$_2$ dissolved in HCl and 0.5 M H$_2$SO$_4$). After incubation for 6 min at 25°C, the absorbance at 660 nm was measured using a Shimadzu UV-1800 spectrometer (Shimadzu Corporation). Data were analyzed using GraphPad Prism (GraphPad Software). In the steady-state kinetics, the data for ATP, L-arginine, and (β-Asp-Arg)$_4$ were fitted with a Michaelis-Menten curve, and the curve fitting for L-aspartate was performed using an allosteric sigmoidal equation as follows:

$$\text{Reaction rate} = \frac{V_{\max}[S]^h}{K_{\text{half}}{}^h + [S]^h} = \frac{V_{\max}[S]^h}{K_{\text{prime}} + [S]^h} \qquad (1)$$

where $V_{\max}$ is the maximum enzyme velocity; $[S]$ is the substrate concentration; $K_{\text{half}}$ is the concentration of substrate that produces a half-maximal enzyme velocity, and $h$ is the Hill coefficient[48].

### Sequence alignments

A homologous sequence search was performed using the amino acid sequences of *Te*CphA1 (Accession No. MBS9770029.1) and GshF from *Pasteurella multocida* (Accession No. WP_010906990.1) in Protein BLAST[49]. CLUSTAL W[50] was used for multiple sequence alignments using default parameters, and the results were displayed by ESPript 3.0[51]. Sequence logos were created using multiple sequence alignment data in the WebLogo server[52].

### Reporting summary

Further information on research design is available in the Nature Research Reporting Summary linked to this article.

## Data availability

Cryo-EM maps generated in this study have been deposited in the Electron Microscopy Data Bank (EMDB) under accession codes EMD-32381 for *Te*CphA1 in the apo state, EMD-32382 for *Te*CphA1 bound with ATPγS, and EMD-32383 ($C_1$) and EMD-32384 ($C_2$) for *Te*CphA1 bound with ATPγS, aspartate, and (β-Asp-Arg)$_4$. The structural coordinates are available in the PDB under accession codes 7WAC for *Te*CphA1 in the apo state, 7WAD for *Te*CphA1 bound with ATPγS, and 7WAE ($C_1$) and 7WAF ($C_2$) for *Te*CphA1 bound with ATPγS, aspartate, and (β-Asp-Arg)$_4$. Enzymatic activity data generated in this study are provided in the Source Data file. The PDB coordinates used in this study are as follows: 7LGJ and 7LGQ. Source data are provided with this paper.

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

## Acknowledgements

We thank Y. Sakamaki and M. Kikkawa (Cryo-EM facility at the University of Tokyo) for their help in cryo-EM data collection and A. Nakamura (Gakusyuin University) for the technical support to build the structural model of $(\beta\text{-Asp-Arg})_4$. This research was supported by the Platform Project for Supporting Drug Discovery and Life Science Research (Basis for Supporting Innovative Drug Discovery and Life Science Research (BINDS)) from AMED under Grant Number JP21am0101077/ JP22ama121010 (T. Miyakawa) and JP21am0101071/JP22ama121001 (*support number 1678*) (T.S.), and the Finance Science and Technology Project of Hainan Province under Grant Number ZDKJ202018.

## Author contributions

M.T. and T.S. supervised the project. T.Miyakawa, J.Y., and M.T. designed the experiments. J.Y., A.F., and Y.M. prepared recombinant proteins. A.F. and Y.M. performed biochemical studies with assistance from T.Miyakawa. T.Miyakawa, N.A., and M.K. collected cryo-EM data. T.Miyakawa and M.K. processed the data with the support of N.A., T.Moriya, and T.Muramatsu. T.Miyakawa and A.F. performed model building and refinement. T.Miyakawa, J.Y., M.K., N.A., A.F., and T.Moriya wrote the manuscript with assistance from all the authors. M.T. and T.S. edited the manuscript.

## Competing interests
