## [Peer Review File · Nature Communications]

Structural bases for aspartate recognition and polymerization efficiency of cyanobacterial cyanophycin synthetaseREVIEWER COMMENTS

Reviewer #1 (Remarks to the Author):

Because I'm not a structural biologist, I will comment only on the cyanophycin synthetase assay, on which I have been specifically asked. The assay used follows the release of phosphate from ATP consumed in the reaction. Phosphate was determined colorimetrically. This is an unusual assay of this enzyme, which has previously been determined mainly by the incorporation of a 3H- or 14C-labeled substrate (aspartate or arginine) into a cyanophycin primer. I suggest that the authors provide as Suppl material a basic characterization of the reaction showing that the determination of cyanophycin synthetase activity by the release of phosphate is working properly. Some attention should also be put on the concentrations used for aspartate (5 mM) and arginine (0.5 mM), which may not be optimal. It would be of interest that the authors show that such concentrations permit an adequate estimation of the reaction.

Other than that, the introduction fails to adequately reflect the importance of cyanophycin in the physiology of N₂ fixation in cyanobacteria. It is stated that cyanophycin has a role in the distribution of fixed N in diel cycles, which is correct for example for *Trichodesmium* (ref. 9). However, cyanophycin is also key for the distribution of fixed N when N₂ fixation and photosynthesis are segregated spatially, as for example in *Anabaena* (ref. 10). This should be clarified. And both *Trichodesmium* and *Anabaena* are filamentous cyanobacteria, i.e., multicellular, not unicellular as generally stated in line 57.

Reviewer #2 (Remarks to the Author):

In "Structural bases for aspartate recognition and polymerization efficiency of cyanobacterial cyanophycin synthetase", Miyakawa et al present a series of cryoEM structures of *Trichodesmium erythraeum* (Te)CphA in complex with substrates: TeCphA alone, TeCphA with ATPγS and TeCphA with ATPγS, (β-Asp-Arg)₄ and Asp. A series of structure-based mutations were made and evaluated by phosphate release assay.

The EM structures appear to be of high quality. There is some question of how faithful the aspartate-bound structure is to the *in vivo* complex, because the 100 mM Asp required to visualize the amino acid is far above the likely intracellular concentration, Asp was visible in only one of four protomers and the reactive atoms are ~6 Å apart (lines 221-227).

The biochemical experiments appear soundly performed but a control without enzyme should be added to fig 2c.

However, the authors must modify the manuscript to make it more clear which observations have been made before. The structure and many of the results presented here are confirmatory of those in reference 13. Figures (in addition to 3d) comparing the two sets of structures should be included to show how similar SuCphA and TeCphA are, and the percent identity and similarity between SuCphA and TeCphA should be stated. In addition, it appears that many of the same mutations were performed in the two studies. The mutations made and the results of mutations should be systematically compared, perhaps in a supplementary table.

There are also several examples of observations analogous to what is known about homologous enzymes, which are stated without citation, such as:

- the interactions with ATP at the G domains (Lines 187-200) being common with ATP-grasp enzymes (and SuCphA),
- the active site and Mlid ATP binding and flexibility of Mlid (Lines 242-262) being common with MurE like enzymes.
- the explanation of lack of poly-Asp polymerase activity being because poly-Asp would lack

interactions of Arg portions (Lines 154-163) was already stated in reference 13

The literature must be cited at the appropriate places, including ref 13 and papers like:

<https://doi.org/10.1002/pro.95>

<https://doi.org/10.1016/j.bioorg.2011.08.004>

<https://doi.org/10.1515/bmc-2013-0024>

<https://doi.org/10.1107/S0907444906038376>

<https://doi.org/10.1111/1462-2920.12171>

<https://doi.org/10.1016/j.jmb.2006.07.066>

When these and the minor points below are addressed, the manuscript will be suitable for publication at a lower profile journal than Nature Communications. Unfortunately, there is not enough novel data to warrant publication in Nat Comm.

Minor points

Line 27-29: "Cyanophycin.. is produced by the nonribosomal peptide synthesis of cyanophycin synthetase (CphA1)": Awkward phrasing, "synthesis of" something usually indicates that something (here, CphA1) is being synthesized. Also, the authors should consider not using the term "non-ribosomal peptide synthesis" because, although correct, it is usually associated with modular NRPS enzymes such as cyclosporin synthetase, to which CphA1 is not related.

Line 55: "making it a nitrogen reservoir for N2 fixation". It is a reservoir of fixed nitrogen, not a reservoir for N2 fixation.

Line 74: "distinct ATP-grasp folds": Are MurE ligases categorized as having ATP-grasp folds?

Line 91-92: "initiation reaction of cyanophycin biosynthesis in the G domain": This reaction occurs repeatedly in cyanophycin synthesis, so initiation may not be the appropriate term here.

Line 127: "which is similar to the structural observation of SuCphA113": Is it also similar to TmCphA and AbCphA?

Line 140: "lariat": The most common biological use of the term lariat is for a splicing biproduct that had a covalent linkage between one end of the strand and a point in its middle. There is no such covalent linkage here and I would advise against using the term.

Line 290: A complete set of substrates would include Arg and a cyanophycin intermediate ending in Asp.

Lines 350-354: "The result of this molecular evolution suggests that Mlid contributes to the linkage reaction between Asp backbones in the G domain." I do not understand what evidence suggests Mlid is involved in G domain - catalyzed Asp ligation. Also consider that ATP-grasp enzymes work well without such a mechanism.

Line 545: Why do the assays contain 10x more Asp than Arg?

We have thoroughly revised our manuscript according to the comments of the reviewers. The following are point-by-point responses to the comments, and the revised portions of the manuscript are marked in red. Page and line numbers are those of the revised manuscript.

Reviewer #1:

[Comment 1]

Because I'm not a structural biologist, I will comment only on the cyanophycin synthetase assay, on which I have been specifically asked. The assay used follows the release of phosphate from ATP consumed in the reaction. Phosphate was determined colorimetrically. This is an unusual assay of this enzyme, which has previously been determined mainly by the incorporation of a 3H- or 14C-labeled substrate (aspartate or arginine) into a cyanophycin primer. I suggest that the authors provide as Suppl material a basic characterization of the reaction showing that the determination of cyanophycin synthetase activity by the release of phosphate is working properly. Some attention should also be put on the concentrations used for aspartate (5 mM) and arginine (0.5 mM), which may not be optimal. It would be of interest that the authors show that such concentrations permit an adequate estimation of the reaction.

[Reply]

We appreciate the important comments on the activity measurements of *TeCphA1*. Following your comments, we performed a steady-state kinetic analysis of *TeCphA1*. As shown in Supplementary Fig. 10a, *TeCphA1* required four substrates, L-aspartate, L-arginine, (β -Asp-Arg)₄, and ATP, for the catalytic reaction. Steady-state kinetic analysis showed a typical Michaelis-Menten kinetics for L-arginine, (β -Asp-Arg)₄, and ATP (Supplementary Fig. 10b). The K_m values for arginine, (β -Asp-Arg)₄, and ATP were 0.0437 ± 0.0018 mM, 0.786 ± 0.043 mM, and 1.87 ± 0.19 mM, respectively. In contrast, aspartate acts as a positive effector of the catalytic reaction (Supplementary Fig. 10b). Curve fitting and kinetic parameter calculation by GraphPad Prism estimated that the Hill coefficient (h), K_{half} , and K_{prime} ($= K_{half}^h$) were 1.92 ± 0.07 , 18.0 ± 1.1 mM, and 254 ± 21 mM, respectively. These results suggested that our enzyme-assay conditions for arginine, (β -Asp-Arg)₄, and ATP (0.5 mM arginine, 2 mM (β -Asp-Arg)₄, and 4 mM ATP) were appropriate. While the concentration of aspartate is rather low for a stable enzyme-activity measurement (Supplementary Fig. 10b), we determined the aspartate concentration at 5 mM by referring to an earlier experiment (ref. 47). In addition, we have corrected the enzyme amounts in the reaction solution because those values were incorrect in the original manuscript.

We have revised the manuscript as follows (revisions are underlined):

- 1) “The reaction mixture contained 2.5–10 μg of purified *TeCphA1*, 50 mM Tris-HCl (pH 8.2), 20 mM KCl, 20 mM MgCl_2 , 1 mM DTT, 4 mM ATP, 5 mM L-aspartate, 0.5 mM L-arginine and 2 mM synthetic cyanophycin primer peptide of different lengths (1- to 5-mer). There are essentially the same conditions as in a previous report⁴⁷ with a small modification (replacement of 10 mM 2-mercaptoethanol with 1 mM DTT). For enzyme kinetics, the activity of *TeCphA1* (1 μM as a tetramer) was measured by varying the concentration of one of the four substrates, ATP, L-aspartate, L-arginine, and $(\beta\text{-Asp-Arg})_4$. Kinetic analysis revealed that our enzyme-assay conditions for arginine, $(\beta\text{-Asp-Arg})_4$, and ATP were appropriate. While the concentration of aspartate is rather low for a stable enzyme-activity measurement, we determined the aspartate concentration at 5 mM by referring to an earlier experiment⁴⁷.” (page 27, line 616–page 28, line 626).
- 2) The methods for the kinetic analysis have been added to the “CphA1 activity assays” section (page 27, line 619–page 28, line 626; and page 28, lines 635–641) with new citations of refs. 47 and 49.
 47. Krehenbrink, M & Steinbüchel, A. Partial purification and characterization of a non-cyanobacterial cyanophycin synthetase from *Acinetobacter calcoaceticus* strain ADP1 with regard to substrate specificity, substrate affinity and binding to cyanophycin. *Microbiology (Reading)* **150**, 2599–2608 (2004).
 49. Copeland, R. A. *Enzymes: A Practical Introduction to Structure, Mechanism, and Data Analysis, 2nd Edition* (Wiley, Hoboken, 2000).
- 3) We have added the following descriptions of the kinetic analysis in Supplementary Fig. 10. “*TeCphA1* required four substrates, L-aspartate, L-arginine, $(\beta\text{-Asp-Arg})_4$, and ATP, for the catalytic reaction (Supplementary Fig. 10a). Steady-state kinetic analysis showed a typical Michaelis-Menten kinetics for L-arginine, $(\beta\text{-Asp-Arg})_4$, and ATP (Supplementary Fig. 10b). In contrast, aspartate acts as a positive effector of the catalytic reaction with the Hill coefficient (h) 1.92 ± 0.07 . In addition, K_{half} and $K_{\text{prime}} (= K_{\text{half}}^h)$ were estimated as 18.0 ± 1.1 mM and 254 ± 21 mM, respectively. As shown in Supplementary Fig. 10b, the catalytic activity of *TeCphA1* increases with the increasing aspartate concentration; the cellular aspartate concentration may be a key regulator for the catalytic activity of *TeCphA1*. The results of the kinetic analysis for aspartate (Supplementary Fig. 10b) suggest that 100 mM aspartate is appropriate for the cryo-EM analysis. While it is unclear why the aspartate molecule was visible in only one subunit, the cryo-EM structure of *TeCphA1* with ATP γ S, $(\beta\text{-Asp-Arg})_4$, and aspartate may be a snapshot of the equilibrium state of complex formation between the enzyme and aspartate.” (page 10, lines 200–212).

Furthermore, the graphs of relative activity of mutants (Figs. 2b and 4b) have been replaced with a

corrected graph due to an error in the control data used in the original manuscript as follows (see below). A “Source Data” file has been provided for the numerical values of the enzymatic activity.

[Comment 2]

Other than that, the introduction fails to adequately reflect the importance of cyanophycin in the physiology of N₂ fixation in cyanobacteria. It is stated that cyanophycin has a role in the distribution of fixed N in diel cycles, which is correct for example for *Trichodesmium* (ref. 9). However, cyanophycin is also key for the distribution of fixed N when N₂ fixation and photosynthesis are segregated spatially, as for example in *Anabaena* (ref. 10). This should be clarified. And both *Trichodesmium* and *Anabaena* are filamentous cyanobacteria, i.e., multicellular, not unicellular as generally stated in line 57.

[Reply]

Thank you for the comment. Following your suggestion, we have clarified the temporal and spatial strategies of cyanobacteria for separating CO₂- and N₂-fixing as follows: “Simultaneous oxygenic photosynthesis and N₂ fixation are a significant challenge for microorganisms because the O₂ produced from CO₂ fixation is inhibitory to nitrogenase, which catalyzes the conversion of N₂ to NH₃⁸. To solve this problem, diazotrophs have developed physical strategies to temporally and spatially separate nitrogenase from O₂. The heterocyst-forming cyanobacteria, such as *Anabaena* spp., perform CO₂ fixation and N₂ fixation in different cells: CO₂ fixation in the vegetative cells and N₂ fixation in heterocysts⁹. Other cyanobacteria, such as *Trichodesmium* spp., temporally segregate the processes by

a diel cycle with CO₂ fixation during the day and N₂ fixation at night¹⁰.” (page 4, lines 59–67).

Reviewer #2

[Comment 1]

In “Structural bases for aspartate recognition and polymerization efficiency of cyanobacterial cyanophycin synthetase”, Miyakawa et al present a series of cryoEM structures of *Trichodesmium erythraeum* (Te)CphA in complex with substrates: TeCphA alone, TeCphA with ATP γ S and TeCphA with ATP γ S, (β -Asp-Arg)₄ and Asp. A series of structure-based mutations were made and evaluated by phosphate release assay.

The EM structures appear to be of high quality. There is some question of how faithful the aspartate-bound structure is to the in vivo complex, because the 100 mM Asp required to visualize the amino acid is far above the likely intracellular concentration, Asp was visible in only one of four protomers and the reactive atoms are ~ 6 Å apart (lines 221-227).

[Reply]

We appreciate the helpful comment. Based on the results of a new kinetic study (Supplementary Fig. 10), we consider that 100 mM aspartate is appropriate to visualize aspartate. As shown in Supplementary Fig. 10b, the enzyme activity continued to increase in the aspartate concentration range of 2.5 to 60 mM, and the catalytic reaction velocity was suggested to peak at around 100 mM Asp. Therefore, an aspartate concentration of 100 mM is appropriate for stable visualization of the equilibrium state of the catalytic reaction of *TeCphA1*.

[Comment 2]

The biochemical experiments appear soundly performed but a control without enzyme should be added to fig 2c.

[Reply]

We have added negative control data (without an enzyme) for each primer peptide (DP = 1–5) to Fig. 2c in the revised manuscript.

[Comment 3]

However, the authors must modify the manuscript to make it more clear which observations have been

made before. The structure and many of the results presented here are confirmatory of those in reference 13. Figures (in addition to 3d) comparing the two sets of structures should be included to show how similar SuCphA and TeCphA are, and the percent identity and similarity between SuCphA and TeCphA should be stated.

[Reply]

We have added a figure showing the structural and sequence similarity of *TeCphA1* to *SuCphA1* as Supplementary Fig. 7. *TeCphA1* has a high sequence identity (69.6%) and similarity (83.2%) to *SuCphA1* in the region consisting of the N, G, and M domains (residues 1–876). We compared structures of *TeCphA1* in the substrate-bound state and *SuCphA1* that bound a cyanophycin analog (β -Asp-Arg)₈-NH₂ and ATP to the G domain (PDB 7LGJ)¹³. These structures are similar to each other. The overall structures of the N domain and each module of the G and M domains (G_{core}, G_{lid}, G_o, M_{core}, and M_{lid}) were quite similar between *TeCphA1* and *SuCphA1* (RMSD = 0.521–1.601 Å). The spatial arrangement of the N domain and the G_{core} and M_{core} modules was also well conserved between *TeCphA1* and *SuCphA1* (Supplementary Fig. 7b), whereas the relative orientation of the other modules versus G_{core} and M_{core} was different (Supplementary Fig. 7c,d). In particular, the distance between the M_{lid} and N domain of *TeCphA1* M_{lid} is shorter than that of *SuCphA1* (Supplementary Fig. 7d). These statements have been added on page 7, line 140–page 8, line 151 in the revised manuscript.

[Comment 4]

In addition, it appears that many of the same mutations were performed in the two studies. The mutations made and the results of mutations should be systematically compared, perhaps in a supplementary table.

[Reply]

Thank you for the comment. Following your suggestion, we attempted to prepare a table to summarize our study and that of Sharon et al. (ref. 13). However, since no numerical data were provided for the *SuCphA1* mutants in the earlier study, it was difficult for us to prepare the comparison table. Therefore, we compared the results of the mutational analyses using a graph of the enzyme activities in the earlier publication. Only three mutations (E215A, H267A, and R309A) of the G domain were commonly analyzed in the two studies. Of the three, R309A and H267A (and their corresponding mutations in *SuCphA1*) showed almost no activities. Both residues seem to be essential for the catalytic activities of *TeCphA1* and *SuCphA1*. On the other hand, while the assay conditions were not the same between the two, the E215A mutation was different between *TeCphA1* and *SuCphA1*. The E215A of *TeCphA1* lost most of the catalytic activity (Fig. 2b), but the corresponding mutation of *SuCphA1* retained nearly half the activity. Since the interaction between E215 and the C-terminal Arg sidechain of cyanophycin contributes more to the cyanophycin binding in *TeCphA1* than in *SuCphA1*, the results of this mutational

analysis seem to be explained by the difference in the structures. The relevant statements have been added on page 9, lines 174–184, and page 12, lines 247–250 in the revised manuscript. Our results for the other new mutations would also support the structural findings for the residues required for the substrate aspartate binding (R323A, N394A, and R458A in Fig. 2b) and the function of M_{lid} (Δ M_{lid}, K499A, F692L, and A755G in Fig. 4b).

[Comment 5]

There are also several examples of observations analogous to what is known about homologous enzymes, which are stated without citation, such as:

- the interactions with ATP at the G domains (Lines 187-200) being common with ATP-grasp enzymes (and SuCphA),
- the active site and M_{lid} ATP binding and flexibility of M_{lid} (Lines 242-262) being common with MurE like enzymes.
- the explanation of lack of poly-Asp polymerase activity being because poly-Asp would lack interactions of Arg portions (Lines 154-163) was already stated in reference 13

The literature must be cited at the appropriate places, including ref 13 and papers like:

<https://doi.org/10.1002/pro.95>

<https://doi.org/10.1016/j.bioorg.2011.08.004>

<https://doi.org/10.1515/bmc-2013-0024>

<https://doi.org/10.1107/S0907444906038376>

<https://doi.org/10.1111/1462-2920.12171>

<https://doi.org/10.1016/j.jmb.2006.07.066>

[Reply]

Thank you for the comment. We are very sorry for the inappropriate citations. We have added suggested references and other related literature.

- the interactions with ATP at the G domains (Lines 187-200) being common with ATP-grasp enzymes (and SuCphA),

The description of the interaction with ATP at the G domain has been changed as follows (revisions are underlined): “The G domain adopts an ATP-grasp fold that requires nucleophilic attack of the C-terminal backbone carboxyl group of cyanophycin on ATP for Asp backbone elongation^{13,19}. However, since these two functional groups are negatively charged, it is necessary to avoid electrostatic repulsion between them. The ATP bound to the ATP-grasp fold typically interacts with one to three Mg²⁺ ions¹⁹. In the G domain of *TeCphA1*, the γ -phosphate chelates two Mg²⁺ ions together with two acidic residues,

Asp431 and Glu450 (Figs. 2a and 3a). This structural feature is similar to *Su*CphA1¹³ and other ATP-grasp enzymes, such as bacterial glutathione synthetases²⁰ and *N*⁵-carboxyaminoimidazole ribonucleotide synthetase (PurK)¹⁹. In addition, the α - and β -phosphates of ATP γ S interact with the positively charged sidechains of Lys261 and Lys220, respectively (Fig. 3a). While the triphosphate of ATP is neutralized overall by these interactions, there are still two conserved charged residues, Arg309 and Arg323, around the γ -phosphate.” (page 11, line 226–page 12, line 238).

The following studies have been cited as refs. 19 and 20.

<https://doi.org/10.1016/j.bioorg.2011.08.004>

<https://doi.org/10.1021/bi9605245>

We have added a description of the P-loop^G contributing to the activity as follows: “This loop corresponds to the B loop of PurT and PurK²¹, which are ATP-grasp enzymes of *E. coli*. These enzymes function in the de novo purine biosynthetic pathway, and the B loop is required to efficiently generate an acylphosphate intermediate in these enzymes. However, His267, which seems to be important to stabilize the acylphosphate intermediate in *Te*CphA1, is not conserved in PurT and PurK²¹.” (page 12, lines 250–255).

The following study has been cited as ref. 21.

<https://doi.org/10.1002/pro.95>

Reference 21 has also been cited in the following sentence: “The C-terminal backbone carboxy group of cyanophycin is activated by ATP to form a high-energy acylphosphate intermediate required for the ATP-grasp enzymes²¹, which is stabilized by His267 on the P-loop^G together with Arg309.” (page 17, lines 359–362).

- the active site and M_{lid} ATP binding and flexibility of M_{lid} (Lines 242-262) being common with MurE like enzymes.

We have added the following sentence to a paragraph describing the flexibility of M_{lid}: “A large shift of M_{lid} was observed in the other CphA1 enzymes¹³ and Mur ligases¹⁵⁻¹⁷.” (page 7, lines 138 and 139).

The following reports have been cited as refs. 15–17.

<https://doi.org/10.1016/j.jmb.2006.07.066>

<https://doi.org/10.1107/S0907444906038376>

<https://doi.org/10.1111/1462-2920.12171>

We have added a description about a common ATP binding mode between M_{lid} and Mur ligases as follows: “These interactions with ATP are highly conserved in the Mur ligases, which synthesize the peptide stem of bacterial peptidoglycan^{22,23}.” (page 14, lines 297 and 298).

The following reports have been cited as refs. 22 and 23.

<https://doi.org/10.1128/JB.185.14.4152-4162.2003>

<https://doi.org/10.1515/bmc-2013-0024>

We have added the following description of the M_{lid} movement upon substrate binding: “In the Mur ligases, the interaction with the substrate induces a rotation of the C-terminal domain, which corresponds to M_{lid} of CphA1 enzymes, resulting in a closed conformation^{15,17}. However, the orientation of M_{lid} is almost unchanged by the substrate (cyanophycin) interaction in *Su*CphA1 (RMSD = 0.158 Å, 2,583 atoms of M_{core} and M_{lid})¹³.” (page 16, lines 336–340).

The following reports have been cited as refs. 15 and 17.

<https://doi.org/10.1016/j.jmb.2006.07.066>

<https://doi.org/10.1111/1462-2920.12171>

- the explanation of lack of poly-Asp polymerase activity being because poly-Asp would lack interactions of Arg portions (Lines 154-163) was already stated in reference 13

As described above in the response to Comment 4, the results of the E215A mutational analysis of *Te*CphA1 and *Su*CphA1 can be explained by the difference in the structures. In particular, we would like to emphasize the importance of the Arg sidechain at the C-terminal residue of cyanophycin. The relevant statements have been changed along with the citation of ref. 13: “Although the importance of the Arg sidechains for the lack of poly-Asp polymerase activity has been explained by the structural observation of *Su*CphA1^{13,18}, our results showed that the Arg sidechain of the 4th unit is a major contributor to $(\beta\text{-Asp-Arg})_4$ recognition and determines the position of the 4th Asp backbone at the active site of the G domain to avoid polymerization progression with the aspartate backbone alone.” (page 9, lines 184–188).

Minor points:

[Comment 6]

Line 27-29: “Cyanophycin.. is produced by the nonribosomal peptide synthesis of cyanophycin synthetase (CphA1)”: Awkward phrasing, “synthesis of” something usually indicates that something

(here, CphA1) is being synthesized. Also, the authors should consider not using the term “non-ribosomal peptide synthesis” because, although correct, it is usually associated with modular NRPS enzymes such as cyclosporin synthetase, to which CphA1 is not related.

[Reply]

Thank you for the comment. We have changed the relevant sentences as follows.

Abstract (page 3, lines 31–33)

“Cyanophycin, a natural biopolymer consisting of equimolar amounts of aspartate and arginine as the backbone and branched sidechain, respectively, is produced by the nonribosomal peptide synthesis of cyanophycin synthetase (CphA1)” has been changed to “Cyanophycin is a natural biopolymer consisting of equimolar amounts of aspartate and arginine as the backbone and branched sidechain, respectively. It is produced by a single enzyme, cyanophycin synthetase (CphA1),”.

Discussion (page 16, lines 343–345)

“Cyanophycin is a unique biopolymer that consists of the β -Asp-Arg dipeptide as a repeating unit (Supplementary Fig. 1) and is produced by CphA1-dependent nonribosomal peptide synthesis.” has been changed to “Cyanophycin is a unique biopolymer that consists of the β -Asp-Arg dipeptide as a repeating unit (Supplementary Fig. 1) and is produced by CphA1-dependent peptide synthesis.”

[Comment 7]

Line 55: “making it a nitrogen reservoir for N₂ fixation”. It is a reservoir of fixed nitrogen, not a reservoir for N₂ fixation.

[Reply]

Thank you for the comment. We have changed the phrase “making it a nitrogen reservoir for N₂ fixation” to “making it a reservoir of fixed nitrogen” (page 4, line 59).

[Comment 8]

Line 74: “distinct ATP-grasp folds”: Are MurE ligases categorized as having ATP-grasp folds?

[Reply]

As described in the suggested study (ref. 15), MurE ligases are not categorized as having ATP-grasp folds. We have changed the passage “Recently, the tertiary structure of CphA1 has become available and reveals that amino acid polymerization is driven by unifying two distinct ATP-grasp folds¹³. CphA1 enzymes are composed of three domains: the N-terminal domain (N domain), the middle

glutathione synthetase-like domain (G domain), and the C-terminal MurE-like muramyl ligase domain (M domain).” to the following: “Recently, the tertiary structure of CphA1 has become available and reveals that amino acid polymerization is driven by three distinct domains¹³: the N-terminal domain (N domain), the middle glutathione synthetase-like domain (G domain) with the ATP-grasp fold, and the MurE-like muramyl ligase domain (M domain).” (page 5, lines 78–82).

[Comment 9]

Line 91-92: “initiation reaction of cyanophycin biosynthesis in the G domain”: This reaction occurs repeatedly in cyanophycin synthesis, so initiation may not be the appropriate term here.

[Reply]

Following your suggestion, we have deleted “initiation” (page 6, lines 95–97).

[Comment 10]

Line 127: “which is similar to the structural observation of SuCphA1¹³”: Is it also similar to TmCphA and AbCphA?

[Reply]

Unfortunately, *TmCphA1* was determined as an apo state, and ATP was observed only in the G domain of *AbCphA1*. To explain the situation, we have revised the manuscript as follows (underlined): “Therefore, M_{core} assumes a distinct catalytic site from the G domain in *TeCphA1*, which is similar to the structural observation of *SuCphA1*¹³.” to “Therefore, M_{core} assumes a distinct catalytic site from the G domain in *TeCphA*. The ATP-bound forms of the G and M domains were also observed in *SuCphA1*¹³. On the other hand, the structural evidence of the ATP-bound forms of the G and M domains for *TmCphA* and *AbCphA* remains insufficient¹³.” (page 7, lines 131–134).

[Comment 11]

Line 140: “lariat”: The most common biological use of the term lariat is for a splicing biproduct that had a covalent linkage between one end of the strand and a point in its middle. There is no such covalent linkage here and I would advise against using the term.

[Reply]

Thank you for the comment. We have changed the phrase “a lariat shape” to “a hook-like shape” according to your comment (page 8, line 160; and page 10, line 193).

[Comment 12]

Line 290: A complete set of substrates would include Arg and a cyanophycin intermediate ending in Asp.

[Reply]

Thank you for the comment. We have changed “a complete set of substrates” to “a complete set of substrates for the reaction of the G domain” (page 16, lines 348 and 349).

[Comment 13]

Lines 350-354: “The result of this molecular evolution suggests that M_{lid} contributes to the linkage reaction between Asp backbones in the G domain.” I do not understand what evidence suggests M_{lid} is involved in G domain - catalyzed Asp ligation. Also consider that ATP-grasp enzymes work well without such a mechanism.

[Reply]

We have changed the relevant sentence to clarify our meaning. The new sentence reads as follows: “CphA2, which synthesizes cyanophycin by directly linking β -Asp-Arg dipeptides, lacks not only the P-loop^M but also the M_{lid} module to abolish the activity for Arg sidechain condensation²⁹, which is supported by our mutational data of *TeCphA1*, which does not exert the CphA2 activity (Fig. 4b and Supplementary Fig. 10a). This molecular evolution may imply that the presence of M_{lid} is unfavorable for the reaction to proceed efficiently only in the G domain, rather than alternately in the two domains.” (page 19, lines 411–416)

[Comment 14]

Line 545: Why do the assays contain 10x more Asp than Arg?

[Reply]

Thank you for the comment. We determined the concentrations of the substrates for the enzyme assay based on the results of the kinetic analysis (Supplementary Fig. 10) and earlier reports (ref. 47). Since K_m for L-arginine was 0.0437 ± 0.0018 mM, 0.5 mM Arg was considered sufficient for the kinetic analysis. On the other hand, L-aspartate acts as a positive effector for the catalytic reaction; the catalytic activity increases with the increasing aspartate concentration (Supplementary Fig. 10b). Therefore, 5 mM aspartate is still rather low for stable enzyme-activity measurement. However, we determined the Asp concentration at 5 mM by referring to the earlier experiment. (Please see also our response to Comment 1 of Reviewer #1.)

The following study has been cited as ref. 47.

<https://doi.org/10.1099/mic.0.27140-0>

Thank you again for your insightful input. Your kind and prompt judgement about this paper would be greatly appreciated.

REVIEWER COMMENTS

Reviewer #1 (Remarks to the Author):

The authors have satisfactorily addressed the issues I raised in my first review.

Reviewer #2 (Remarks to the Author):

Miyakawa et al have performed steady-state kinetic analyses of TeCphA1 and submitted a revised manuscript. The manuscript is improved.

The authors should comment on their observation that ATP has a single K_m . ATP is used in for two different reactions in two different active sites and a previous study reported significantly different K_m values for ATP in these two active sites (reference 47).

I do suggest that a kinetics expert should be consulted to verify that the kinetic values calculated justify the inclusion of 100 mM aspartate in the co-complex sample preparation. I am not a kinetics expert, and it still appears very strange to me that 100 mM Asp is needed to see Asp bound at one of the four G domain active sites. I do not know the intracellular concentration of Asp in cyanobacterial cells, but *E. coli* cells are thought to have ~4.2 mM Asp (Bennett et al, Nature Chemical Biology, 2009), and *E. coli* harbouring CphA1 can make large quantities of cyanophycin.

Reviewer #1:

[Overall comments]

The authors have satisfactorily addressed the issues I raised in my first review.

[Reply]

We thank you again for your constructive comments regarding the first revision of our manuscript.

Reviewer #2

[Overall comments]

Miyakawa et al have performed steady-state kinetic analyses of TeCphA1 and submitted a revised manuscript. The manuscript is improved.

[Reply]

We greatly appreciate the further helpful comments to improve our manuscript. We have revised our manuscript according to your comments. The following are point-by-point responses to the comments, and the revised portions of the manuscript are marked in red. Page and line numbers are those of the revised manuscript.

[Comment 1]

The authors should comment on their observation that ATP has a single K_m . ATP is used in for two different reactions in two different active sites and a previous study reported significantly different K_m values for ATP in these two active sites (reference 47).

[Reply]

Thank you for the comment. As the reviewer pointed out, Krehenbrink et al. (ref. 47 that has been cited as ref. 19 in the revised manuscript) separately determined the K_m values for ATP in two different active sites by detecting only the condensation reaction of either L-aspartate or L-arginine using radioisotope-labeled substrates. However, we measured the apparent K_m value for ATP for the sequential condensation reaction of TeCphA1 by fitting the Michaelis-Menten equation. The apparent K_m value for such a sequential reaction was also reported in ref. 47, and they concluded that the reaction

was rate-limited primarily by the site for L-aspartate condensation reaction with lower affinity for ATP. Based on these results reported in ref. 47, we should have stated the possibility that the K_m values of *TeCphA1* for ATP are different between the G and M domains and the sequential condensation reaction of *TeCphA1* is rate-limited primarily by the site with lower affinity for ATP. We have added the following description: “While we have determined the apparent K_m value for ATP, *TeCphA1* has two active sites in the G and M domains for the different catalytic reactions with ATP. As observed for the other CphA1 enzyme¹⁹, the sequential condensation reaction of *TeCphA1* may be rate-limited primarily by the site with lower affinity for ATP when the K_m values for ATP differ between the G and M domains.” (page 10, lines 203–207).

[Comment 2]

I do suggest that a kinetics expert should be consulted to verify that the kinetic values calculated justify the inclusion of 100 mM aspartate in the co-complex sample preparation. I am not a kinetics expert, and it still appears very strange to me that 100 mM Asp is needed to see Asp bound at one of the four G domain active sites. I do not know the intracellular concentration of Asp in cyanobacterial cells, but *E. coli* cells are thought to have ~4.2 mM Asp (Bennett et al, Nature Chemical Biology, 2009), and *E. coli* harbouring CphA1 can make large quantities of cyanophycin.

[Reply]

Thank you for the comment. First, it is usual to use a high substrate concentration when obtaining the 3D structure of the enzyme-substrate (or its analog) complex because it is necessary to increase the occupancy of the substrate in the enzyme. We have typically used substrate (or analog) concentrations at least 5 times higher than K_m when determining the enzyme-substrate complex. The optimal conditions of the substrate for the structural analysis can be significantly different from those derived from kinetic analysis. However, as the reviewer suggested, it is unclear why the aspartate molecule was visible in only one subunit even at 100 mM L-aspartate. We discussed the reason with the authors; some are familiar with enzyme kinetics.

There are several possibilities. First, substrates/analogs (ATP γ S, (β -Asp-Arg)₄, and aspartate) in the sample buffer solution may not be appropriate for forming the enzyme-aspartate complex in all tetrameric protomers. Since *TeCphA1* requires multiple substrates for its complete reaction, several intermediate states seem to exist in the catalytic reaction cycle. Since we have not completely understood the reaction cycle, there might be an unknown process or factor to increase the affinity for aspartate. Second, an allosteric mechanism might affect the affinity for aspartate. In this case, inter-subunit interactions could affect the affinity for the aspartate of one subunit. Unfortunately, we could not find inter-subunit interactions that affect the affinity for aspartate in this study.

Anyway, since aspartate was visible only in one subunit, it is reasonable to consider that our sample conditions are not appropriate to bind the aspartate to (all) protomers. There is a possibility, however, that the high concentration of aspartate (100 mM) forced the aspartate to become visible while the conditions are not suitable for the aspartate binding.

Finally, we would like to note that enzymes can work in cells under non-optimal conditions. Even under non-optimal conditions, the enzymes play physiological roles. While 4–5 mM aspartate seems not to be biochemically optimal conditions for the *TeCphA1*, *TeCphA1* may be able to synthesize cyanophycin in cells. It is also possible that local concentrations of aspartate in cells are significantly higher than the average value. For example, cyanophycin granules may provide a reaction compartment where enzymes and substrates are concentrated, as observed in liquid-liquid phase separation (Peeples & Rosen, 2021).

Peeples, W & Rosen, M. K. Mechanistic dissection of increased enzymatic rate in a phase-separated compartment. *Nat. Chem. Biol.* 17, 693–702 (2021).

<https://doi.org/10.1038/s41589-021-00801-x>

Unfortunately, our kinetic and biological analyses on *TeCphA1* are not enough yet to clearly answer this question, as stated above (only the apparent K_m has been obtained as described above). We need more kinetic and *in vivo* analyses to clarify this point. We will address them in future work.